# High-resolution analysis of human centromeric chromatin

Daniël P Melters[1] , Minh Bui[1], Tatini Rakshit[1,2], Sergei A Grigoryev[3] , David Sturgill[1,4], Yamini Dalal[1]

**Centromeres are marked by the centromere-specific histone H3 variant CENP-A/CENH3. Throughout the cell cycle, the constitutive centromere-associated network is bound to CENP-A chromatin, but how this protein network modifies CENP-A nucleosome conformations in vivo is unknown. Here, we purify endogenous centromeric chromatin associated with the CENP-C complex across the cell cycle and analyze the structures by single-molecule imaging and biochemical assays. CENP-C complex–bound chromatin was refractory to MNase digestion. The CENP-C complex increased in height throughout the cell cycle culminating in mitosis, and the smaller CENP-C complex corresponds to the dimensions of in vitro reconstituted constitutive centromere-associated network. In addition, we found two distinct CENP-A nucleosomal configurations; the taller variant was associated with the CENP-C complex. Finally, CENP-A mutants partially corrected CENP-C overexpression–induced centromeric transcription and mitotic defects. In all, our data support a working model in which CENP-C is critical in regulating centromere homeostasis by supporting a unique higher order structure of centromeric chromatin and altering the accessibility of the centromeric chromatin fiber for transcriptional machinery.**

## Introduction

The kinetochore is a large proteinaceous complex, which physically connects centromeric chromatin to the mitotic microtubule spindles. Inaccuracies in kinetochore assembly can lead to the formation of dicentric chromosomes, or chromosomes lacking kinetochores. In either case, chromosomes fail to segregate faithfully, which drives genomic instability (Paul et al, 2022; Hosea et al, 2024). Electron microscopy studies of mitotic centromeres reveal a two-layered electron-dense structure that is over 200 nm in width and over 50 nm in depth (Robbins & Gonatas, 1964; Brinkley & Stubblefield, 1966; Comings & Okada, 1970; Rattner et al, 1975; Esponda, 1978; McEwen et al, 1998), delineated into the inner and outer kinetochore.

At the base of the inner kinetochore sits the histone H3 variant CENP-A (CENH3), which provides the epigenetic and structural foundation to assemble the kinetochore, recruiting inner kinetochore proteins CENP-C and CENP-N in human, fruit fly, and chicken cells (Régnier et al, 2005; Carroll et al, 2009, 2010; Mendiburo et al, 2011; Allu et al, 2019). These kinetochore proteins recruit the other constitutive centromere-associated network (CCAN)/inner kinetochore proteins (Klare et al, 2015; Walstein et al, 2021). Together, these inner kinetochore components provide recognition motifs for outer kinetochore proteins in the conserved KMN network (Cheeseman et al, 2006; Przewloka et al, 2007; DeLuca & Musacchio, 2012; Weir et al, 2016).

Despite major advances in understanding the hierarchy of the centromere and kinetochore proteins (Milks et al, 2009; Cheeseman, 2014; Klare et al, 2015; McKinley & Cheeseman, 2016a) and in vitro reconstituted CCAN structures (Weir et al, 2016; Pesenti et al, 2018, 2022; Hinshaw & Harrison, 2019; Yan et al, 2019; Walstein et al, 2021; Tian et al, 2022; Yatskevich et al, 2022; Dendooven et al, 2023), little is known about the in vivo physical features of the inner kinetochore bound to centromeric chromatin in multicellular eukaryotes. The only example to date is the mitotic kinetochores isolated from budding yeast using a FLAG-tagged outer kinetochore component Dsn1 as bait, yet the structure of the inner kinetochore remains unclear (Gonen et al, 2012). Point centromeres of budding yeast are unique because they display a one-to-one correspondence, in which a single CENP-A nucleosome (Bloom & Carbon, 1982; Saunders et al, 1988; Furuyama & Henikoff, 2009; Kingston et al, 2011; Krassovsky et al, 2012; Furuyama et al, 2013; Henikoff et al, 2014; Díaz-Ingelmo et al, 2015) binds to a single microtubule via the kinetochore. In contrast, the human centromeres are regional centromeres comprised of megabase-sized α-satellite DNA (Waye & Willard, 1989; Rudd et al, 2006). Super-resolution microscopy suggests that each human centromere harbors ~400 CENP-A molecules (Bodor et al, 2014), which eventually associate with ~17 mitotic microtubule spindles (Suzuki et al, 2015). Unfolding of chicken chromatin using low-salt buffers indicated that centromeric chromatin is folded into a boustrophedon (Ribeiro et al, 2010; Vargiu et al, 2017), whereas a looping model has been suggested for the point centromere (Lawrimore & Bloom, 2019). What the endogenous kinetochore structure or structures are in human cells remains unknown.

Here, we report that we were able to purify and analyze human CENP-C complexes bound to centromeric chromatin, using nanoscale, immunofluorescence imaging, and biochemical approaches. Native chromatin immunoprecipitation (nChIP) of the

[1]National Cancer Institute, Center for Cancer Research, Laboratory of Receptor Biology and Gene Expression, Bethesda, MD, USA   [2]Department of Chemistry, Shiv Nadar Institution of Eminence, Delhi, India   [3]Pennsylvania State University, College of Medicine, Hershey, PA, USA   [4]National Cancer Institute, Center for Cancer Genomics, Bethesda, MD, USA

Correspondence: dalaly@mail.nih.gov; daniel.melters@nih.gov

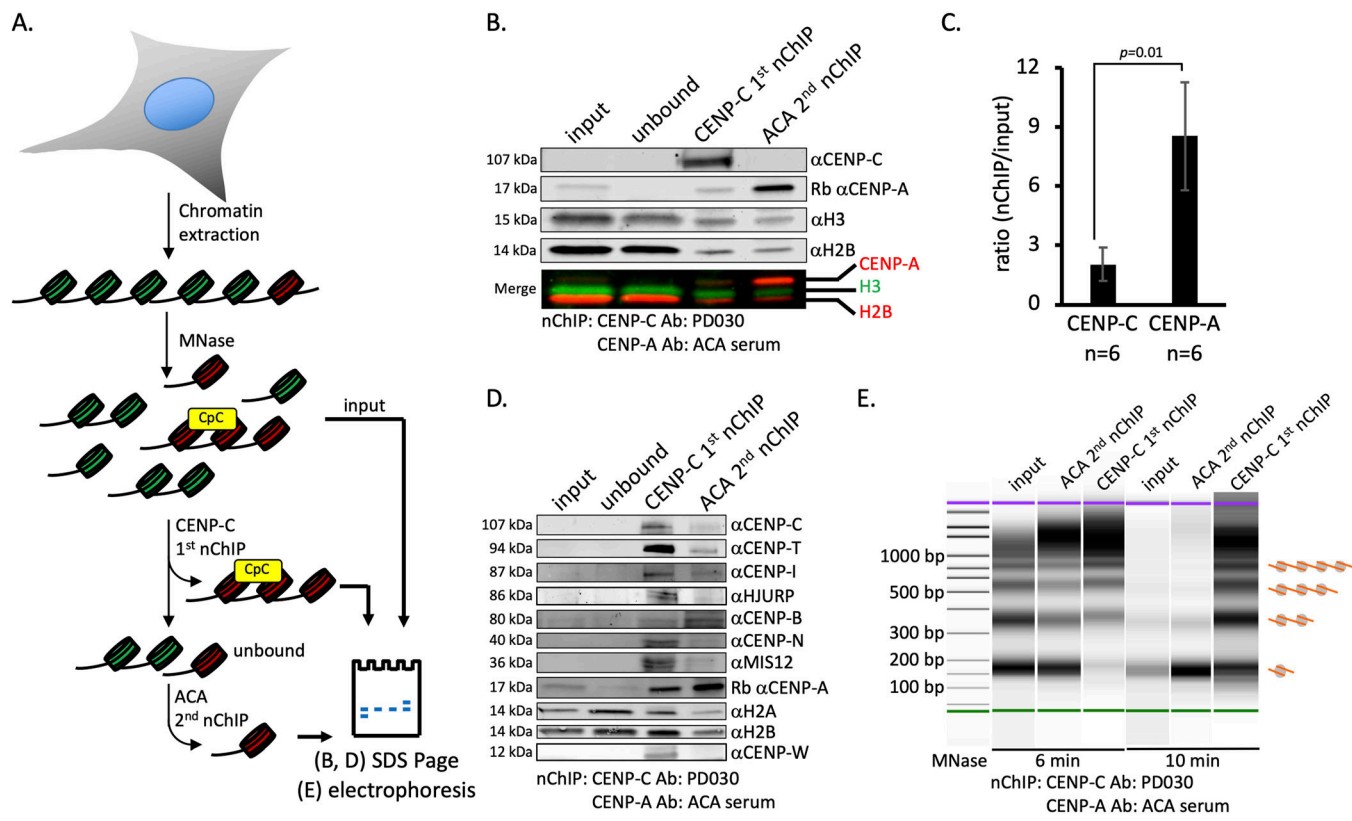

**Figure 1. CENP-C nChIP pulls down a subfraction of CENP-A and kinetochore proteins and is refractory to MNase digestion.**
Chromatin was extracted from HeLa cells after 6-min MNase digestion, followed by serial nChIP. The first CENP-C nChIP was done, followed by CENP-A nChIP on the unbound fraction from the first nChIP. **(A)** Schematic representation of the experimental setup for serial native ChIP of CENP-C followed by ACA (blue = H3 nucleosome; red = CENP-A nucleosome; yellow = CENP-C complex). **(B)** Representative Western blot of serial CENP-C (CENP-A first nChIP) and ACA nChIP (CENP-A second nChIP). n = number of experiments quantified. **(C)** Quantification of nChIP–Western blot (paired $t$ test, significance was determined at $P < 0.05$; error bars = SD). **(D)** Representative Western blot analysis of serial CENP-C and ACA nChIP, stained for inner and outer kinetochore proteins. **(E)** DNA fragment analyzed by BioAnalyzer after MNase digestion for six or 10 min.
Source data are available for this figure.

inner kinetochore protein CENP-C brought down a large complex bound to CENP-A chromatin that was more refractory to nuclease digestion than canonical or CENP-A–only chromatin. The CENP-C complex increased in height throughout the cell cycle culminating in mitosis, whereas Feret's diameter remained the same. Furthermore, using two different CENP-C and CENP-A antibodies, two distinct CENP-A populations were identified, one pulled down by CENP-C and the other by subsequent CENP-A nChIP. These two CENP-A nucleosome populations were physically distinct from one another. To test the potential role of these two CENP-A forms, we overexpressed CENP-C in cells, which negatively impacted centromeric α-satellite transcription and led to extensive mitotic defects. CENP-A mutants, either one that binds CENP-C to a nucleosome with an H3 histone fold domain, or one that cannot bind CENP-C, rescued centromeric transcription and partially rescued mitotic defects. Altogether, these data support a working model in which CENP-C is critical in regulating centromere homeostasis by supporting a unique higher order structure of centromeric chromatin and altering the accessibility of the centromeric chromatin fiber for transcriptional machinery.

# Results

## CENP-C binds a subfraction of CENP-A chromatin

In previous work (Dimitriadis et al, 2010; Bui et al, 2012), we noticed a small fraction of macromolecular complexes within CENP-A nChIP that were large, compacted, and refractory to standard nucleosome analysis (Dimitriadis et al, 2010; Bui et al, 2012). We were curious to examine these larger complexes. As CENP-A is bound by the CCAN, we predicted that pulling down a CCAN component will enrich for this large, compacted structure. Two of the CCAN proteins, CENP-C and CENP-N, bind CENP-A nucleosomes directly (Régnier et al, 2005; Carroll et al, 2009, 2010; Mendiburo et al, 2011; Allu et al, 2019), and CENP-C promotes targeting of other CCAN proteins (Klare et al, 2015). Therefore, we developed a gentle serial nChIP assay to purify CENP-C bound to chromatin from asynchronous HeLa cells (Fig 1A). We moderately digested nuclear chromatin with micrococcal nuclease (MNase) resulting in a range of nucleosome arrays (Rakshit et al, 2020). Using a commercial polyclonal anti-CENP-C antibody (PD030), we pulled down CENP-C, as well as the centromeric chromatin mark CENP-A and nucleosome mark H2B (Fig 1B). To test

whether CENP-C pulled down all CENP-A molecules, we nChIPed the unbound fraction of the first CENP-C nChIP with ACA serum, which contains antibodies against CENP-A, CENP-B, and CENP-C (Fig 1A) (Moroi et al, 1980; Earnshaw & Rothfield, 1985). We did not observe CENP-C that came down in the second ACA nChIP (Fig 1B). In contrast, CENP-A did come down with the first CENP-C nChIP and the second ACA nChIP (Fig 1B). Interestingly, a substantial amount of CENP-A came down in the second ACA nChIP compared with the first CENP-C nChIP (Fig 1C). These results in asynchronous cells were compared with our previous findings in cells synchronized to early G1 (Melters et al, 2019, 2023). The current results indicate that CENP-C does not associate with all available CENP-A nucleosomes.

### CENP-C nChIP pulls down CCAN kinetochore components

Throughout the cell cycle, the CCAN is bound to centromeric chromatin (McKinley & Cheeseman, 2016b; Pesenti et al, 2016), so we first focused on asynchronous HeLa cells. To test whether anti-CENP-C nChIP pulls down other kinetochore proteins besides CENP-C, we performed Western blot analyses. In addition to histones CENP-A, H2A, and H2B, CCAN components CENP-I, CENP-N, CENP-W, and CENP-T were enriched in CENP-C nChIP (Fig 1D). CENP-B is a centromeric protein that binds directly to DNA via the CENP-B box (Earnshaw et al, 1987; Masumoto et al, 1989; Yoda et al, 1996) and should be pulled down by ACA nChIP. Indeed, we find that CENP-B is pulled down by both the first CENP-C and the second ACA nChIP (Fig 1D). Furthermore, we show that CENP-C, but not ACA nChIP, pulled down the CENP-A chaperone HJURP (Dunleavy et al, 2009; Foltz et al, 2009; Bergmann et al, 2011) (Fig 1D). Overall, these results indicate that purified CENP-C bound to chromatin contains CCAN components, and by extension potentially the entire CCAN.

### CENP-C complexes bind to MNase refractory centromeric DNA

H3 nucleosomes protect 147 bp of DNA upon MNase digestion (Kornberg, 1974; Luger et al, 1997), whereas CENP-A protects 120 bp of DNA (Tachiwana et al, 2011; Bui et al, 2012). Both CENP-C and the CCAN have been shown to bind to DNA in vitro (Milks et al, 2009; McKinley & Cheeseman, 2016b; Pesenti et al, 2016), which made us wonder what DNA fragment sizes were protected by CENP-C nChIP in vivo. To assess the chromatin array size, we used high-resolution capillary electrophoresis. Bulk chromatin, CENP-A chromatin, and CENP-C–associated chromatin were mildly digested with MNase for either 6 or 10 min, and DNA was purified. Both bulk chromatin and serial ACA nChIP showed a very similar MNase ladder, especially after 10 min of MNase digestion. This highlights that CENP-A forms regular nucleosome arrays in vivo. In contrast, CENP-C nChIP showed noticeably longer chromatin arrays (Fig 1E). These results indicate that CENP-C is associated with DNA that is less accessible to MNase digestion. This observation was further supported by a sedimentation assay, using a 5–20% glycerol gradient, that showed that CENP-C–associated chromatin had a distinct sedimentation pattern compared with serial ACA nChIP (Fig S1).

To test whether CENP-C–associated chromatin occupied unique centromeric sequences compared with CENP-C–depleted CENP-A chromatin, we performed ChIP-seq. DNA that came down with CENP-C and ACA nChIP was sequenced using PacBio long-read technology, and obtained sequence reads were aligned to α-satellite monomers (see the Methods and Material section). ChIP-seq analysis showed that both CENP-C and ACA nChIP pulled down similar centromeric sequences (Fig S2).

Together, these descriptive analyses showed that in vivo CENP-C is associated with centromeric CENP-A chromatin and forms a complex that is refractory to MNase digestion and likely represents the inner kinetochore/CCAN.

### CENP-C nChIP comes down as a large complex

Next, we wanted to understand the physical dimensions of CENP-C bound to centromeric chromatin. Therefore, we extracted chromatin from asynchronous HeLa cells and performed serial CENP-C and ACA nChIP (Fig 2A). After a gentle purification method, we split the samples in half and imaged the same samples independently (by two different operators) using two complementary high-resolution single-molecule methods: transmission electron microscopy (TEM) and in-air tapping-mode atomic force microscopy (AFM) (Figs 2B–D and S3). To retain integrity of the chromatin, samples were analyzed within 24 h after purification. AFM is a topographical imaging technology where samples deposited on an atomically flat surface (mica) are scanned using a tip with a diameter of 3–7 nm. This allows us to measure the dimensions of samples at sub-nanometer resolution. CENP-C nChIP samples were isolated and without fixatives were deposited onto functionalized mica and subsequently imaged by AFM. We observed large polygonal structures (height: 5.8 ± 2.1 nm; diameter: 27.4 ± 2.7 nm, Table 1) with a roughly circular footprint (0.79 ± 0.11, Table 1), which were generally associated with four to six nucleosomes that could be visually identified (Figs 2C and D and S3). The same samples (but fixed with 0.005% glutaraldehyde) were analyzed in parallel by TEM and displayed similar features (Fig 2B), including the association of two to four nucleosomes at the periphery of the complex and exposing nucleosome-free DNA partially hidden by bulky electron density. Interestingly, by AFM, we observed that the largest of CENP-C complexes was associated with 156 ± 90 bp long nucleosome-free DNA (Fig 1D). Thus, CENP-C nChIP pulls down a distinctive large polygonal structure that is associated with chromatin in which CENP-C in complex with nucleosomes forms a unique higher order structure.

### Development of immuno-AFM to verify the identity of nucleosomes associated with the CENP-C complex

To confirm the identity of the nucleosomes that were associated with endogenous CENP-C complexes, we developed a single-molecule–based method. Inspired by classical immuno-EM protocols (Dimitriadis et al, 2010) and recognition AFM (Wang et al, 2008), we adapted immunolabeling for in-air tapping-mode AFM (immuno-AFM).

We first visualized samples by AFM either without antibody or with primary (1°) mouse monoclonal anti-CENP-A antibody or with 1° CENP-A antibody; and either secondary (2°) anti-mouse antibody or 2° anti-mouse Fab fragment. The 1° antibody alone was 0.8 ± 0.2 nm in height, and the addition of the 2° antibody resulted in a height increase to 2.0 ± 0.5 nm (Fig 3A and B). To confirm the 1°

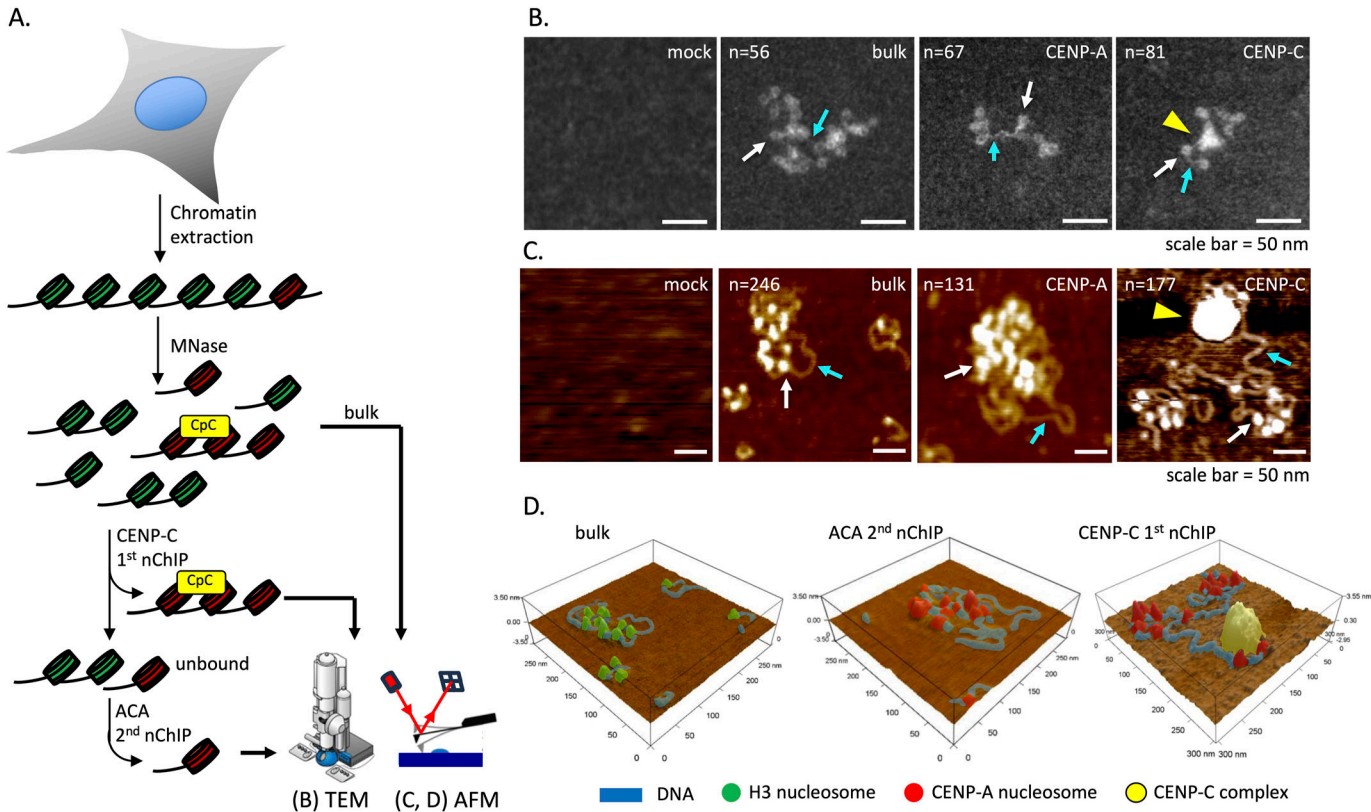

**Figure 2. Anti-CENP-C antibody pulls down a large polygonal complex.**
**(A)** Chromatin was extracted from HeLa cells after 6-min MNase digestion, followed by either mock, ACA, or CENP-C nChIP. Unbound chromatin was used for bulk chromatin. **(B, C)** Representative AFM (B) and TEM (C) images of either mock, bulk chromatin, CENP-A chromatin, or CENP-C chromatin showed that the CENP-C complex is a large polygonal structure with excessive nucleosome-free DNA. White arrows point to nucleosomes. Blue arrows point to naked DNA. The yellow arrowhead points to the CENP-C complex. **(D)** 3D rendering of the bulk, CENP-A chromatin, and CENP-C complex associated with chromatin. The large polygonal structure (yellow) is shown associated with DNA (blue) and nucleosomes (red for CENP-A, green for H3). n = number of particles measured.
Source data are available for this figure.

antibody's specificity, we used in vitro reconstituted recombinant H3 or CENP-A nucleosomes as before and incubated them with either no antibody (no Ab), 1° alone, or 1° + 2° antibodies, respectively. As expected, control in vitro reconstituted H3 nucleosomes did not show a shift in particle height in the presence of anti-CENP-A antibodies (no Ab: 2.2 ± 0.2 nm; 1°: 2.1 ± 0.2 nm; and 2°: 2.2 ± 0.1 nm, resp.). However, in vitro reconstituted CENP-A nucleosomes did increase in height upon the addition of 1° and 2° antibodies (no Ab: 2.2 ± 0.2 nm; 1°: 2.5 ± 0.3 nm; and 4.6 ± 1.4 nm with 2° antibody or 3.2 ± 0.6 nm with Fab fragment, resp., Figs 3A and B and S4).

Having standardized this approach on in vitro reconstituted samples, we next applied this method to in vivo samples, namely, endogenous bulk chromatin and CENP-C purified chromatin (Fig 3A and B). Similar to reconstituted H3 nucleosomes, endogenous bulk chromatin did not demonstrate a shift in particle height when incubated with anti-CENP-A antibodies (no Ab: 2.3 ± 0.2 nm; 1°: 2.4 ± 0.3 nm; and 2°: 2.3 ± 0.1 nm, resp.). In contrast, nucleosomal particles that came down in the CENP-C nChIP displayed a shift in height when challenged with anti-CENP-A antibodies (no Ab: 2.4 ± 0.4 nm; 1°: 2.6 ± 0.4 nm; and 2°: 3.9 ± 1.3 nm, resp.). These results support the notion that CENP-A nucleosomes are indeed associated with the purified CENP-C complexes.

**CENP-A nucleosomes associated with CENP-C are physically distinct**

We next turned our attention to the dimensions of individual nucleosomes. In previous work, we showed that recombinant octameric CENP-A nucleosomes reconstituted on 601 or α-satellite DNA under near-physiological conditions (150 mM NaCl, 2 mM $Mg^{2+}$) behave similar to H3 when measured by in-air tapping-mode AFM, with an average height of ~2.6 nm and a diameter of ~12 nm (Walkiewicz et al, 2014b; Athwal et al, 2015). In contrast, endogenous CENP-A nucleosomes purified from either fruit fly or human cell lines generally display smaller dimensions compared with endogenous H3 nucleosomes, except during the S phase (Dalal et al, 2007; Dimitriadis et al, 2010; Bui et al, 2012, 2013; Athwal et al, 2015). We recapitulated these observations here. Relative to H3 nucleosomes (2.5 ± 0.3 nm), endogenous CENP-A nucleosomes were shorter with an average height of 1.8 ± 0.3 nm (Fig 3C, Table 1).

Subsequently, we examined nucleosomes that are physically associated with the CENP-C complexes (Fig 3C). Nucleosomes associated with CENP-C complexes had distinctly larger dimensions, with a height of 2.5 ± 0.4 nm, compared with CENP-A nucleosomes alone (2.5 ± 0.4 nm versus 1.8 ± 0.3 nm, resp.; Table 1, one-way ANOVA, $P < 0.0001$).

**Table 1. Summary of nucleosome and CENP-C complex AFM measurements.**

| Cell cycle stage | Sample | Antibody | n | Height (nm) | Diameter (nm) | Area (nm²) | Roundness |
|---|---|---|---|---|---|---|---|
| Asynchronous | Bulk nucleosomes | | 167 | 2.7 ± 0.4 | 12.7 ± 0.7 | 130 ± 40 | 0.87 ± 0.05 |
| | CENP-A nucleosomes | ACA | 151 | 1.8 ± 0.3 | 12.4 ± 1.6 | 123 ± 33 | 0.78 ± 0.09 |
| | CENP-C–associated nucleosomes | PD030 | 194 | 2.5 ± 0.4 | 12.8 ± 1.9 | 131 ± 39 | 0.87 ± 0.05 |
| | CENP-C complex | PD030 | 177 | 5.8 ± 2.1 | 27.4 ± 2.7* | 629 ± 358 | 0.79 ± 0.11 |
| G1 | Bulk nucleosomes | | 664 | 2.5 ± 0.4 | 14.0 ± 2.8 | 141 ± 55 | 0.68 ± 0.13 |
| | CENP-A nucleosomes | Custom | 135 | 1.9 ± 0.3 | 14.0 ± 2.0 | 165 ± 47 | 0.77 ± 0.12 |
| | CENP-C–associated nucleosomes | EPR15939 | 405 | 2.4 ± 0.4 | 14.3 ± 2.9 | 154 ± 62 | 0.70 ± 0.13 |
| | CENP-C complex | EPR15939 | 114 | 6.2 ± 2.7 | 64.1 ± 25.8* | 2254 ± 2218 | 0.71 ± 0.13 |
| S | Bulk nucleosomes | | 616 | 2.8 ± 0.7 | 13.5 ± 2.6 | 126 ± 48 | 0.69 ± 0.13 |
| | CENP-A nucleosomes | Custom | 23 | 2.4 ± 0.3 | 13.8 ± 1.9 | 146 ± 44 | 0.61 ± 0.10 |
| | CENP-C–associated nucleosomes | EPR15939 | 220 | 2.5 ± 0.3 | 14.1 ± 2.6 | 141 ± 55 | 0.75 ± 0.12 |
| | CENP-C complex | EPR15939 | 288 | 6.0 ± 2.1 | 67.8 ± 33.4* | 2963 ± 2873 | 0.75 ± 0.12 |
| G2 | Bulk nucleosomes | | 278 | 2.6 ± 0.3 | 14.2 ± 2.7 | 151 ± 56 | 0.73 ± 0.11 |
| | CENP-A nucleosomes | Custom | 96 | 2.0 ± 0.2 | 14.0 ± 2.1 | 162 ± 46 | 0.75 ± 0.11 |
| | CENP-C–associated nucleosomes | EPR15939 | 646 | 2.9 ± 0.2 | 14.4 ± 2.9 | 159 ± 63 | 0.77 ± 0.12 |
| | CENP-C complex | EPR15939 | 45 | 6.7 ± 2.6 | 72.4 ± 25.5* | 2953 ± 2307 | 0.74 ± 0.14 |
| M | Bulk nucleosomes | | 430 | 2.5 ± 0.5 | 14.2 ± 2.8 | 145 ± 55 | 0.69 ± 0.13 |
| | CENP-A nucleosomes | Custom | 824 | 1.9 ± 0.2 | 14.5 ± 1.0 | 179 ± 24 | 0.75 ± 0.10 |
| | CENP-C–associated nucleosomes | EPR15939 | 288 | 2.4 ± 0.6 | 14.0 ± 2.8 | 147 ± 58 | 0.69 ± 0.13 |
| | CENP-C complex | EPR15939 | 196 | 7.5 ± 3.8 | 62.2 ± 26.0* | 2146 ± 1874 | 0.73 ± 0.12 |
| | In vitro H3 nucleosomes | | 175 | 2.5 ± 0.4 | 11.9 ± 1.6 | 90 ± 21 | 0.71 ± 0.12 |
| | HA-H3$^{CpA\ CTD}$ nucleosomes | sc-805 | 118 | 2.6 ± 0.4 | 12.5 ± 1.8 | 104 ± 33 | 0.67 ± 0.13 |
| | HA-CpA$^{\Delta CTD}$ nucleosomes | sc-805 | 110 | 2.1 ± 0.4 | 12.8 ± 2.1 | 111 ± 38 | 0.68 ± 0.13 |

n = number of nucleosomes or complexes measured. Bulk nucleosomes = nucleosomes extracted from HeLa cells; CENP-A nucleosomes = nucleosomes that came down with CENP-A nChIP CENP-A; CENP-C–associated nucleosomes = nucleosomes that came down with CENP-C nChIP; CENP-C complex = complexes that came down with CENP-C nChIP. * = Feret's diameter (nm).

## CENP-C complex associates with physically distinct CENP-A nucleosomes across the cell cycle

Next, we wondered whether the CENP-C complex associated with physically distinct CENP-A nucleosomes displayed similar dynamics across the cell cycle as "bulk" CENP-A nucleosomes do (Bui et al, 2012). As the first commercial polyclonal anti-CENP-C antibody was discontinued, we had to resort to a new anti-CENP-C antibody. We found a second anti-CENP-C monoclonal antibody (ERP15939) that pulled down CENP-C effectively, and the material was useful for both Western blot and AFM analyses (Fig 4). As ACA serum contains antibodies against CENP-A, CENP-B, and CENP-C (Moroi et al, 1980; Earnshaw & Rothfield, 1985), we wanted a more precise antibody to pull down CENP-A. We therefore used a custom CENP-A antibody (Arunkumar et al, 2022; Bui et al, 2024) for the serial second nChIP (Fig 4A). We performed the serial nChIP for HeLa cells synchronized for S, G2, M, and G1 phases after a mild MNase digestion. In all four cell stages, CENP-C effectively pulled down CENP-A and H2B (Fig 4B). The serial CENP-A nChIP pulled down a very small amount of remnant CENP-C in S, G2, and M phases. These results with the second combination of CENP-C/A antibodies confirm that CENP-C binds a subfraction of CENP-A nucleosomes in a cell cycle–independent manner.

In asynchronous cells, we observed that CENP-C–associated CENP-A nucleosomes were taller than the serial nChIPed CENP-A nucleosomes (Fig 3C). Previously, we reported that the structure of CENP-A nucleosomes is dynamic throughout the cell cycle, with CENP-A nucleosomes being only as tall as bulk H3 nucleosomes in the S phase. In all other cell cycle stages, CENP-A nucleosomes were shorter than bulk H3 nucleosomes (Bui et al, 2012). We therefore wondered whether CENP-A nucleosomes associated with the CENP-C complex had similar cell cycle–dependent dynamics. By AFM, we measured the height of bulk H3 nucleosomes and nucleosomes that came down with CENP-C and CENP-A nChIP. We recapitulated our previous findings (Bui et al, 2012) for the second CENP-A nChIP nucleosomes across the cell cycle (G1: 1.9 ± 0.3 nm; S: 2.4 ± 0.3 nm; G2: 2.0 ± 0.2 nm; and M: 1.9 ± 0.2 nm, resp; Fig 4C, Table 1). Yet, CENP-A nucleosomes that came down with the first CENP-C nChIP did not display a cell cycle–dependent structural dynamic (G1: 2.4 ± 0.4 nm; S: 2.5 ± 0.3 nm; G2: 2.9 ± 0.2 nm; and M: 2.5 ± 0.6 nm, resp; Fig 4C, Table 1). Instead, the nucleosomes associated with CENP-C were consistently taller in all four cell cycles (Fig 4C, Table 1).

Altogether, these results point to the presence of two physically distinct CENP-A nucleosomes within the human centromere: one species of CENP-A nucleosome, which is physically shorter, and another

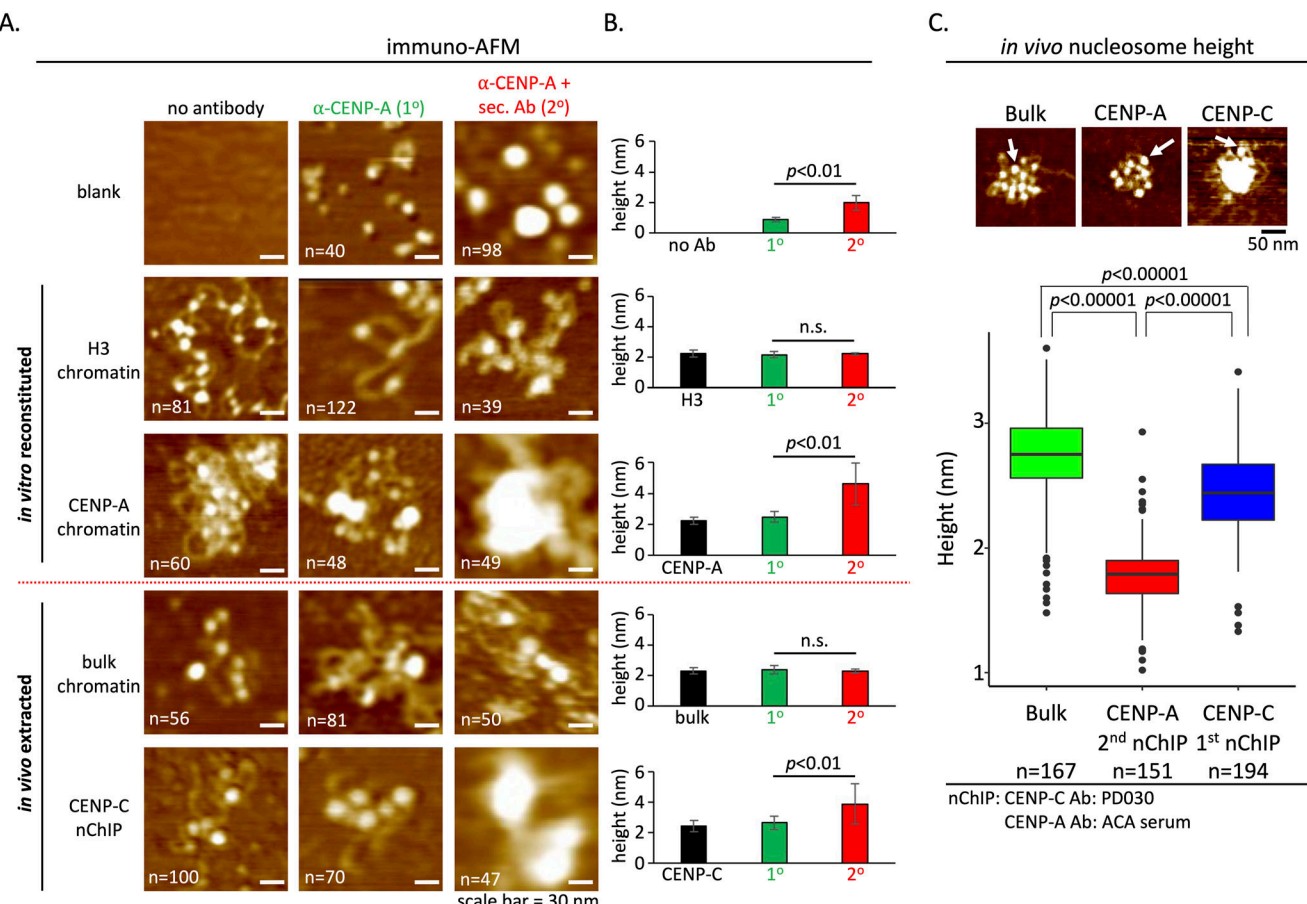

**Figure 3. Immuno-AFM confirms CENP-C associates with CENP-A nucleosomes.**
(A) Representative images for three conditions (no antibody, 1° antibody, and 1° plus 2° antibodies) per sample. The scale bar is 30 nm. (B) Representative AFM images of bulk chromatin, CENP-A nChIP, and CENP-C nChIP are shown with a white arrow pointing to individually identifiable nucleosomes that could be confidently measured. Scale bar = 50 nm. Height measurement of all three conditions is plotted per sample, showing that monoclonal anti-CENP-A antibody only recognized in vitro reconstituted CENP-A nucleosomes and nucleosomes associated with CENP-C, confirming that CENP-C indeed strongly associated with CENP-A nucleosomes (one-way ANOVA, $P <$ 0.01). (C) Nucleosomal height was determined for bulk chromatin, CENP-A nucleosomes were pulled down by CENP-A nChIP, and CENP-A nucleosomes were pulled down by CENP-C nChIP (one-way ANOVA; significance was determined at $P < 0.05$). Arrow points at clearly identifiable nucleosomes that were measured. n = number of particles measured from three independent experiments.
Source data are available for this figure.

species of CENP-A nucleosome associated with the CENP-C complex, which adopts a taller configuration and is stable across the cell cycle.

## CENP-C complex dimensions across the cell cycle

Whereas CENP-A nucleosomes display cell cycle–dependent dynamics, the CCAN is associated with centromeric chromatin throughout the cell cycle (McKinley & Cheeseman, 2016b; Pesenti et al, 2016), with the full kinetochore only needed in mitosis (Dong & Li, 2022). Yet, the physical features of CCAN associated with centromeric chromatin throughout the cell cycle remain untested. Recent cryo-EM work of in vitro reconstituted CCAN provides us with dimensions (Weir et al, 2016; Pesenti et al, 2018, 2022; Hinshaw & Harrison, 2019; Yan et al, 2019; Walstein et al, 2021; Tian et al, 2022; Yatskevich et al, 2022; Dendooven et al, 2023). We predicted that under AFM conditions, the CCAN structure (PDB: 7QOO [Pesenti et al, 2022]) will predominantly fall on its largest surface, just like nucleosomes (Melters et al, 2019). In this scenario, the x-axis and y-axis dimension of the CCAN would be ~15 by

~17 nm (Fig 5A), making an oval shape with maximum Feret's diameter (hereon referred to as Feret's diameter) of ~22.6 nm. In this configuration, the CCAN structure would have a height of 9.3 nm by cryo-EM (Fig 5A). From high-resolution imaging of in-air tapping-mode AFM, we and others noted that nucleosomes appear shorter in height compared with their cryo-EM and crystallography structures by ~50% (~5–5.5 nm versus ~2.5–3 nm, resp.). Extrapolating these observations to the CCAN structure, we predict that by AFM, the CCAN's height would measure around 4.65 nm. The large polygonal CENP-C complex (Fig 5B) that came down with CENP-C nChIP had Feret's diameter of ~48 nm, and a height of ~5.8 nm with a near-circular footprint (Fig 5B, Table 1). These measurements are substantially larger than the CCAN cryo-EM structural dimensions. It has to be noted that the cryo-EM CCAN (Fig 5A) was in vitro reconstituted at a molar ratio of 1:1. This implies that the CENP-C complex associated with centromeric chromatin is more complex than the in vitro reconstituted CCAN version.

One possible reason could be that CENP-C forms a homodimer in vivo (Kingston et al, 2011; Hara et al, 2023) and thus can associate with

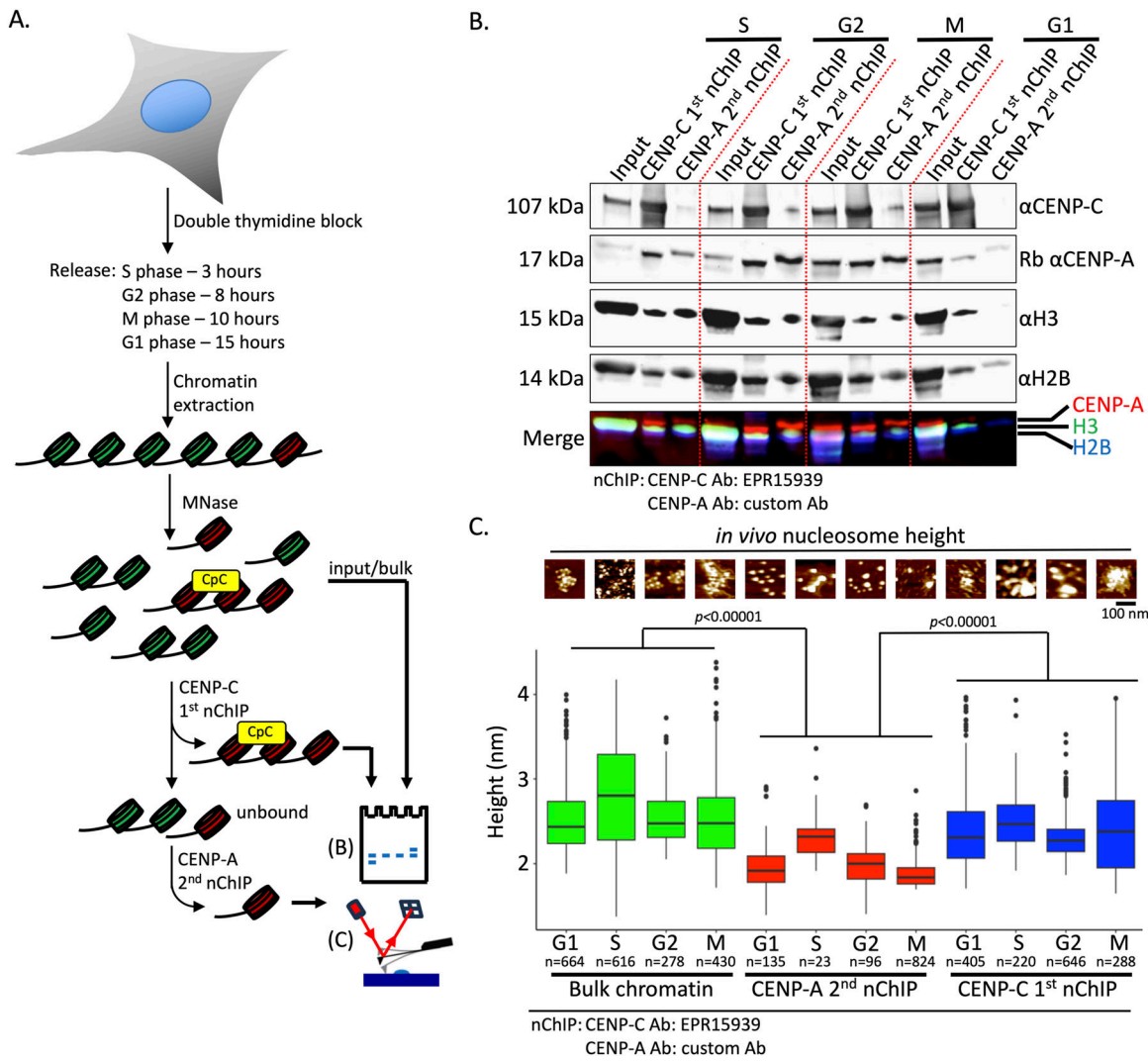

**Figure 4.  CENP-A nucleosomes associated with CENP-C complexes are physically distinct from bulk CENP-A nucleosomes throughout the cell cycle.**
HeLa cells were synchronized for S, G2, M, and G1 phases by a double thymidine block. Chromatin was extracted after a 6-min MNase digestion, and the first CENP-C nChIP was performed. The CENP-A nChIP was done on the unbound fraction from the first nChIP (see Fig 1A). **(A)** Western blot of serial nChIP (first CENP-C nChIP, followed by second CENP-A nChIP). **(B)** Nucleosomal height was determined by AFM from chromatin extracted at the various cell cycle stages. All three samples (bulk chromatin, CENP-A nChIP, and CENP-C nChIP) showed similar heights within samples, but not between samples. The only exception is the increased nucleosomal height of CENP-A nChIP in the S phase. (Tukey's honestly significant difference was performed, and $P < 0.05$ was considered significant.) n = number of particles measured from three independent experiments. Source data are available for this figure.

up to four CENP-A nucleosomes. In chicken DT40 and human cells, CENP-C and the CCAN were found to have a dynamic organization (Hemmerich et al, 2008; Nagpal et al, 2015). To test whether the CENP-C complex structure is dynamic across the cell cycle in human cells, we pulled down the CENP-C complex across the cell cycle (Figs 4C and 5C and S5). First, we compared the height of the CENP-C complexes over the cell cycle and found that the CENP-C complex was tallest in mitosis with a gradual but non-significant increase in G2 compared with G1 and S phases (G1: 6.2 ± 2.7 nm; S: 6.0 ± 2.1; G2: 6.7 ± 2.6 nm; and M: 7.5 ± 3.8 nm, resp.; Fig 5D, Table 1). A Tukey multiple pairwise comparison of the mean showed that the CENP-C complex height was significantly taller in the M phase compared with G1 and S phases. On the other hand, Feret's diameter of the CENP-C complexes remained unchanged over the cell cycle (G1: 64.1 ± 25.8; S: 67.8 ± 33.4; G2: 72.4 ± 25.5; and M:

62.2 ± 26.0, resp.; Fig 5E, Table 1). We also measured the roundness of the CENP-C complex across the cell cycle. We found that the CENP-C complex was less round in G1 compared with the other cell cycle stages (Fig 5F, Table 1). Finally, we tested the correlation between the height and Feret's diameter of the CENP-C complex across the cell cycle (Fig 5G). We found the highest correlation for the S phase ($R^2 = 0.36$) and the lowest correlation for the M phase ($R^2 = 0.15$). It has to be noted that the precise composition of each individual CENP-C complex we imaged is unknown. Previous mass spec analysis of tagged CENP-A nucleosomes showed that both kinetochore and non-kinetochore proteins co-immunoprecipitated (Dunleavy et al, 2009; Foltz et al, 2009; Roulland et al, 2016; Remnant et al, 2019). Therefore, we cannot exclude the possibility that the correlation we found here is in part driven by accumulation of kinetochore and non-kinetochore proteins to the

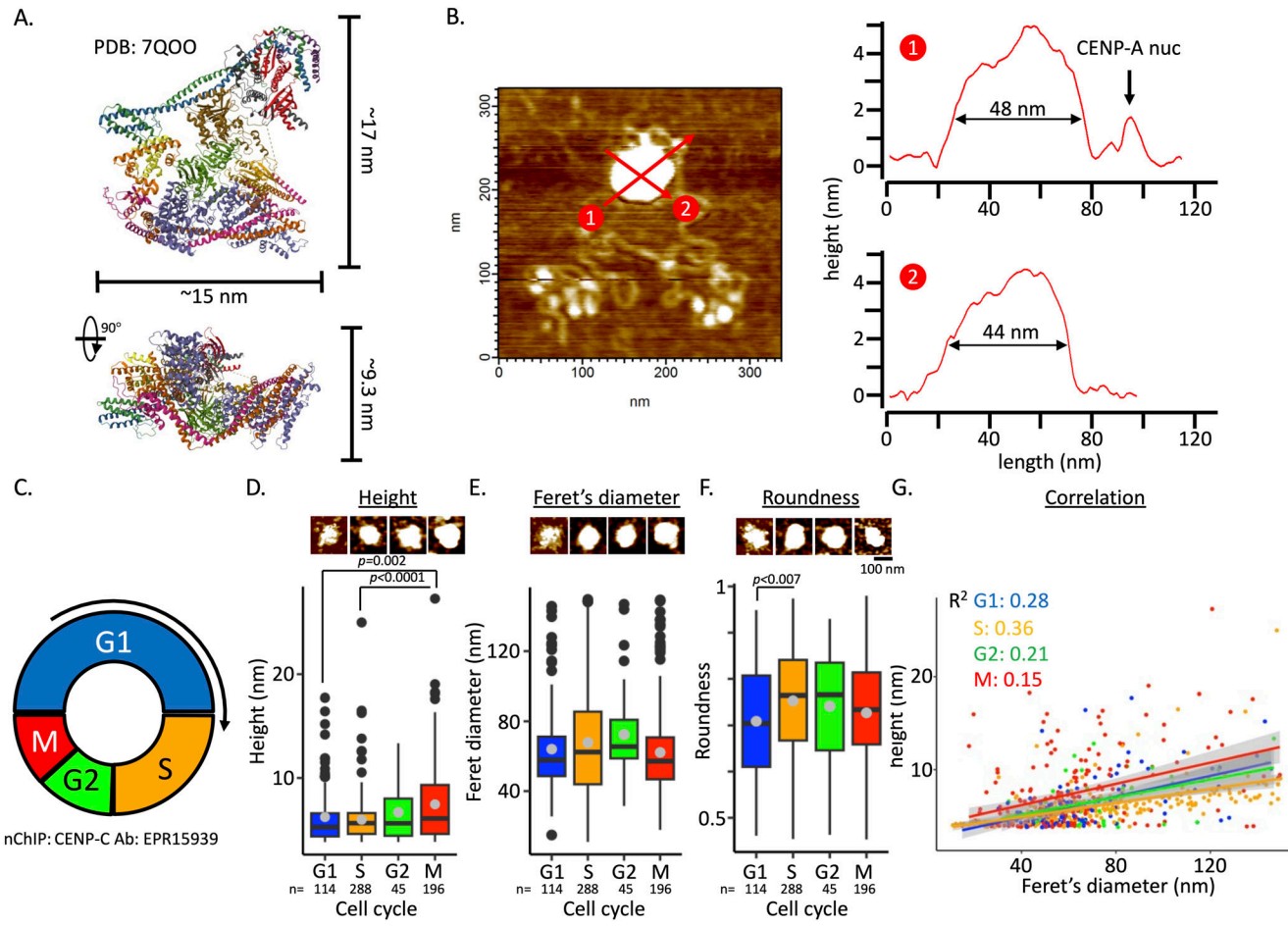

**Figure 5. CENP-C complexes increase in height throughout the cell cycle.**
**(A)** Dimensions from the cryo-EM–derived in vitro reconstituted CCAN structure (PDB: 7QOO [Pesenti et al, 2022]). **(B)** Top view of an AFM image of a CENP-C complex associated with chromatin including stretches of naked DNA, including two cross-sectional profiles of the CENP-C complex showing the near-symmetrical shape of the CENP-C complex. One CENP-A nucleosome can be seen in the #1 cross section. **(C)** Graphical representation of the cell cycle, which is color-coded (blue = G1; orange = S; green = G2; red = M). **(D)** Boxplot of the distribution of the height of the CENP-C complex across the cell cycle. The gray circle represents the mean. (Tukey's HSD was performed, and $P < 0.05$ was considered significant.) n = number of particles measured from three independent experiments. **(E)** Boxplot of the distribution of Feret's diameter of the CENP-C complex across the cell cycle. The gray circle represents the mean. (Tukey's HSD was performed, and $P < 0.05$ was considered significant.) n = number of particles measured. **(F)** Boxplot of the distribution of the roundness factor of the CENP-C complex across the cell cycle. By measuring two lines drawn perpendicular through an object is the roundness factor, with a perfect circle being 1 and an oval being less than 1. The gray circle represents the mean. (Tukey's HSD was performed, and $P < 0.05$ was considered significant.) n = number of particles measured. **(G)** Scatterplot of CENP-C complex height versus CENP-C complex Feret's diameter, showing a comparable correlation across the cell cycle. Each dot represents measurements from a single CENP-C complex.
Source data are available for this figure.

CENP-C complex at various stages in the cell cycle. Nevertheless, our data do indicate that the CENP-C complex undergoes conformational changes as it goes through the cell cycle.

## CENP-A mutants colocalize to centromeres but do not co-immunoprecipitate with endogenous CENP-A

We were curious why CENP-C associates with a subfraction of the available CENP-A nucleosomes in HeLa cells. CENP-C functionally binds to the C-terminal tail domain (CTD) of CENP-A nucleosomes (Carroll et al, 2010; Kato et al, 2013; Falk et al, 2015, 2016; Guo et al, 2017; Ali-Ahmad et al, 2019). Hence, we reasoned that introducing CENP-A mutants would alter the binding landscape of CENP-C. We expressed a fusion of either (1) H3 with the C-terminal tail of CENP-A

(H3$^{CENP-A\ CTD}$), which can bind CENP-C without a centromere localization domain (CATD) (Kato et al, 2013), or (2) CENP-A without its C-terminal tail (CENP-A$^{\Delta CTD}$), which therefore can still be deposited to centromeres by the CENP-A chaperone HJURP, but cannot bind CENP-C (Fig 6A). First, we tested whether the two mutants formed nucleosome-sized particles in HeLa cells. We performed HA nChIP identical to nChIP described above, followed by AFM and Western blot analysis. We used in vitro reconstituted H3 nucleosomes as a control. Similar to what we observed for bulk H3 nucleosomes (Figs 3C and 4C, Table 1), in vitro reconstituted H3 nucleosomes and nChIPed HA-H3-$^{CENP-A\ CTD}$ were similar in height (2.5 ± 0.4 nm and 2.6 ± 0.4 nm, resp.; Fig 6B, Table 1), whereas HA-CENP-A$^{\Delta CTD}$ was shorter (2.1 ± 0.4 nm; Fig 6B, Table 1) similar to what we observed for CENP-A nucleosomes in asynchronous cells and in most cell cycle stages (Figs 3C and 4B).

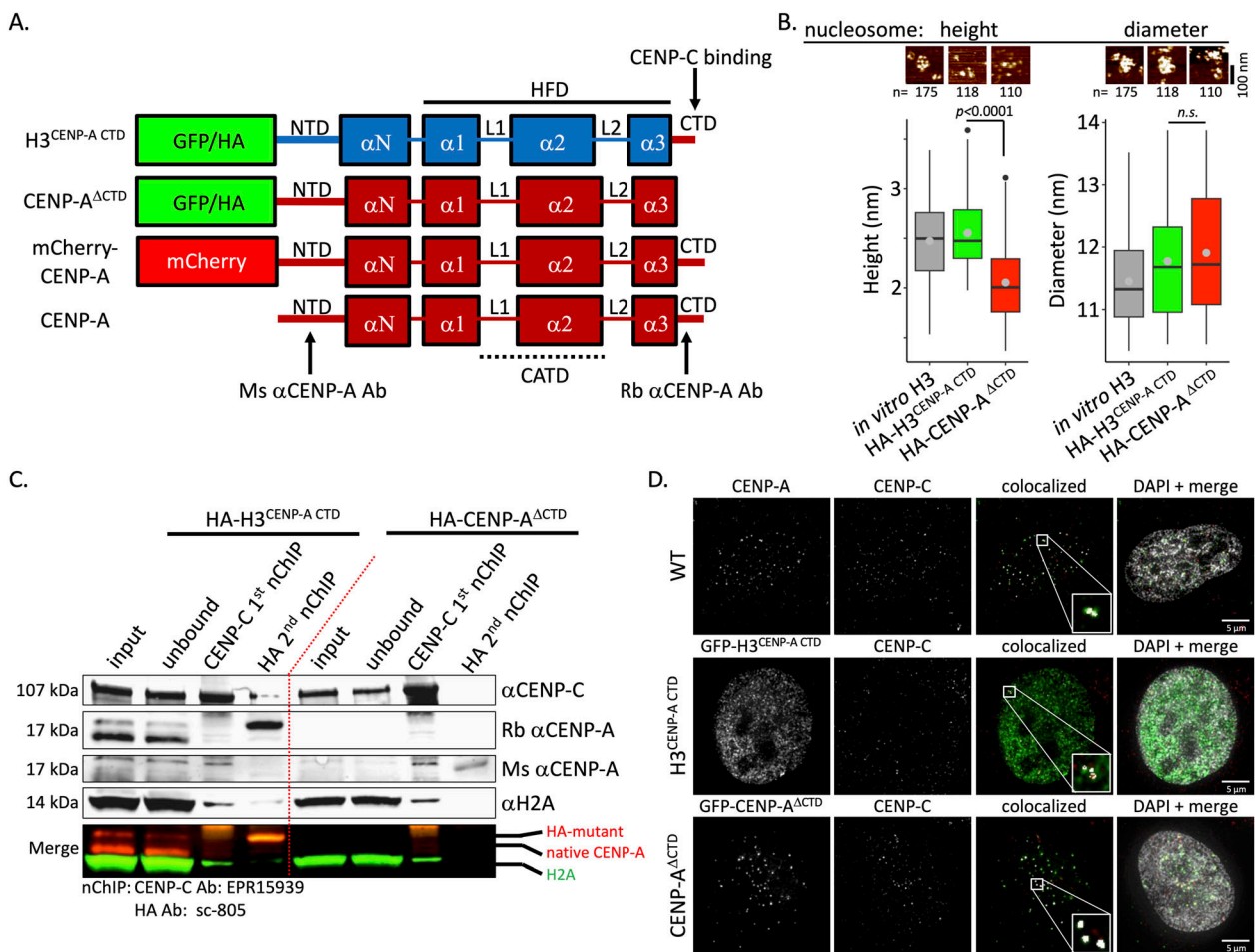

**Figure 6. CENP-A mutants colocalize to centromeres, but do not co-IP with endogenous CENP-A.**
To be able to modulate the two CENP-A populations, we reasoned that the CENP-C levels at the centromere could be modulated by the expression of a CENP-A mutant, namely, histone H3 with the C-terminal tail (CTD) of CENP-A (H3$^{CENP-A\ CTD}$) or CENP-A lacking its C-terminal tail (CENP-A$^{\Delta CTD}$). CATD = centromere targeting domain. HFD = histone fold domain. NTD = N-terminal tail domain. **(A)** Graphical representation of the two CENP-A mutants, indicating where CENP-C binds to CENP-A, and where the epitopes of the two CENP-A antibodies (mouse and rabbit) locate on CENP-A. **(B)** After AFM imaging, nucleosomal height and diameter were determined of the mutant, using in vitro reconstituted H3 nucleosomes as a control. n = number of particles measured. **(C)** To test whether the CENP-A mutants bind endogenous CENP-A or CENP-C, we performed SDS–PAGE analysis for CENP-C–bound chromatin (CENP-C first nChIP) and HA-tagged CENP-A mutants (HA second nChIP). **(D)** To determine centromeric localization, we imaged HeLa cells transfected with GFP-H3$^{CENP-A\ CTD}$ or GFP-CENP-A$^{\Delta CTD}$ and stained for GFP (green) and endogenous CENP-C (centromeric marker; red). Using ImageJ colocalization feature, localization of GFP and CENP-C staining was determined (see inset; white = colocalization). DAPI is in gray. Scale bar = 5 μm. Source data are available for this figure.

Next, we set out to test whether the CENP-A mutants associate with CENP-C and endogenous CENP-A. After serial nChIP (first nChIP = CENP-C; second nChIP = HA), we performed Western blot analysis. No appreciable amount of HA-H3$^{CENP-A\ CTD}$ or HA-CENP-A$^{\Delta CTD}$ came down with the first CENP-C nChIP. In the second HA nChIP, neither CENP-A mutant effectively pulled down CENP-C, although small amounts of CENP-C did come down with the second HA-H3$^{CENP-A\ CTD}$ nChIP. We had to use two CENP-A antibodies to visualize either HA-H3$^{CENP-A\ CTD}$ or HA-CENP-A$^{\Delta CTD}$. The rabbit anti-CENP-A antibody targets the CTD of CENP-A (Fig 6A) and thus could only bind to endogenous CENP-A and HA-H3$^{CENP-A\ CTD}$, but not HA-CENP-A$^{\Delta CTD}$. In contrast, the mouse anti-CENP-A antibody targets the N-terminal domain (NTD) of CENP-A and thus could only bind to endogenous CENP-A and HA-CENP-A$^{\Delta CTD}$ (Fig 6A and C). In all cases, no endogenous CENP-A came down with either CENP-A mutant.

Interestingly, in the presence of HA-CENP-A$^{\Delta CTD}$, we observed very low levels of endogenous CENP-A with either CENP-A antibody (Fig 6C).

Immunofluorescence of the CENP-A mutants (GFP-tagged) in combination with mCherry-CENP-A showed colocalization between the CENP-A mutants and mCherry-CENP-A. As HA-H3$^{CENP-A\ CTD}$ does not have a centromere localization domain, H3$^{CENP-A\ CTD}$ was also found throughout the nucleus as one would expect from H3 nucleosomes (Fig S6). Furthermore, by immunofluorescence, we show that both GFP-tagged CENP-A mutants (GFP-H3$^{CENP-A\ CTD}$ and GFP-CENP-A$^{\Delta CTD}$; Fig 6A) colocalized with endogenous CENP-C (Fig 6D).

Taking these data together, both CENP-A mutants appear to form nucleosomes in vivo and colocalize with the centromere marker mCherry-CENP-A and endogenous CENP-C (Fig 6B and D), yet minimally interact with endogenous CENP-A or CENP-C (Fig 6C).

## CENP-A mutants restore centromeric transcription that is suppressed by CENP-C overexpression

When CID (fruit fly CENP-A), dCENP-C, and CAL1 (fruit fly CID chaperone) were introduced into human cells, they formed a positive feedback loop to maintain a CID on human chromosomes (Roure et al, 2019). Furthermore, previously we postulated that a critical balance might exist between CENP-C–associated CENP-A chromatin and CENP-C–unbound CENP-A chromatin, based on the biophysical features of CENP-A that changed when bound to CENP-C, affecting CENP-A chromatin compaction, centromeric RNAP2 levels, and loading of new CENP-A nucleosomes (Melters et al, 2019, 2023). We therefore speculated whether the CENP-A mutants HA-H3$^{CENP-A\ CTD}$ and HA-CENP-A$^{\Delta CTD}$ could serve as a regulator of excess CENP-C (Fig S7).

First, we performed Western blot analysis after serial nChIP for CENP-C and CENP-A in wild-type (WT) HeLa cells, cells overexpressing CENP-C (CENP-C OE; CENP-C overexpression = ~twofold over endogenous levels), cells overexpressing CENP-C in the presence of HA-H3$^{CENP-A\ CTD}$, and cells overexpressing CENP-C in the presence of HA-CENP-A$^{\Delta CTD}$. When CENP-C was overexpressed, either alone or in the presence of CENP-A mutants, CENP-C came down in both CENP-C and CENP-A nChIP (Fig 7A). This implies that the first CENP-C nChIP was not able to pull down all available CENP-C or not all CENP-C was accessible to the CENP-C antibody. Just as under endogenous CENP-C levels, in the CENP-C overexpression background, both CENP-A mutants did not come down with the first CENP-C nChIP, but only with the second CENP-A nChIP (Fig 7A). Interestingly, under endogenous CENP-C levels, HA-CENP-A$^{\Delta CTD}$ came down in the second CENP-A nChIP with only very low levels of endogenous CENP-A being detected (Fig 6C). Yet, in the CENP-C overexpression background, endogenous CENP-A was detected, but HA-CENP-A$^{\Delta CTD}$ only came down in the second CENP-A nChIP (Fig 7A). In addition, excess CENP-C came down in all three CENP-C overexpression conditions (Fig 7A). These results indicate that the major CENP-C fraction does not associate with either CENP-A mutant.

Previously, we showed that CENP-C overexpression resulted in repression of centromeric α-satellite transcription (Melters et al, 2023). Therefore, we tested whether either or both CENP-A mutants impacted centromeric α-satellite transcription. We normalized the α-satellite transcription to GAPDH transcription. By quantitative PCR, we observed that CENP-C overexpression resulted in a ~60% reduction in α-satellite transcription (Fig 7B), which is in agreement with our previous observation (Melters et al, 2023). When either H3$^{CENP-A\ CTD}$ or CENP-A$^{\Delta CTD}$ was co-expressed in the CENP-C overexpression background, we found that α-satellite transcript levels were very similar to WT conditions (Fig 7B). These results indicate that both CENP-A mutants restored the transcriptional potential of centromeric α-satellite DNA.

## CENP-A mutants partially rescue mitotic defects caused by CENP-C overexpression

Finally, we tested whether CENP-C overexpression impacted mitotic fidelity. Similar to a previous report in chicken DT-40 cells (Fukagawa et al, 1999), the overexpression of CENP-C in HeLa cells resulted in an increase in mitotic defects (40% normal, 30% lagging chromosomes, 16% multipolar spindles, 14% multipolar spindles + lagging chromosomes, Fig 7C and D) relative to WT cells (74% normal, 21% lagging chromosomes, 4% multipolar spindles, 1%

multipolar spindles + lagging chromosomes, Fig 7C and D). The most noticeable mitotic defects were lagging chromosomes and multipolar spindles (Fig 7C and D). When we introduced the CENP-A mutants H3$^{CENP-A\ CTD}$ or CENP-A$^{\Delta CTD}$ mutants into HeLa cells that overexpress CENP-C, we observed that only multipolar spindle defects (with or without lagging chromosomes) were rescued by both H3$^{CENP-A\ CTD}$ (58% normal, 37% lagging chromosomes, 3% multipolar spindles, 2% multipolar spindles + lagging chromosomes, Fig 7C and D) and CENP-A$^{\Delta CTD}$ (65% normal, 27% lagging chromosomes, 6% multipolar spindles, 2% multipolar spindles + lagging chromosomes, Fig 7C and D).

Together, these data indicate that CENP-C overexpression results in an increase in mitotic defects (Fig 7C and D). Introducing either CENP-A mutant (CENP-A$^{\Delta CTD}$ or H3$^{CENP-A\ CTD}$) in the CENP-C overexpression background resulted in a rescue of multipolar spindles (Fig 7C and D).

# Discussion

Here, we describe the structure of purified human CENP-C complexes bound to centromeric chromatin analyzed biochemically, by AFM, immuno-AFM, and TEM. The general architecture reveals that the CENP-C complex is a large polygonal structure that is physically associated with four to six CENP-A nucleosomes (Figs 2, 3, and 5). Biochemical analyses showed that various CCAN components come down with CENP-C nChIP and a large chromatin array (Figs 1 and S1). When we measured the CENP-C complex across the cell cycle, we found a marked increase in height of the CENP-C complex in mitosis compared with G1 and S phases (Fig 5D), whereas Feret's diameter remained unchanged (Fig 5E). Recent cryo-EM work by various groups revealed the structures of in vitro reconstituted budding yeast and human inner kinetochore (Weir et al, 2016; Pesenti et al, 2018, 2022; Hinshaw & Harrison, 2019; Yan et al, 2019; Walstein et al, 2021; Tian et al, 2022; Yatskevich et al, 2022; Dendooven et al, 2023) and human outer kinetochore (Polley et al, 2024; Yatskevich et al, 2024). There is a global consensus on the base structure of the human CCAN, including its direct interaction with naked DNA. Although the relative position and structure of most of the CCAN proteins was uncovered, CENP-C remains structurally a mystery. Only one study used full-length CENP-C in their CCAN reconstitution (Walstein et al, 2021), whereas most studies used a fragment of CENP-C (Weir et al, 2016; Pesenti et al, 2018; Tian et al, 2022; Yatskevich et al, 2022). In all cases, only small fragments of CENP-C showed interpretable densities (Weir et al, 2016; Pesenti et al, 2018, 2022; Hinshaw & Harrison, 2019; Yan et al, 2019; Walstein et al, 2021; Tian et al, 2022; Yatskevich et al, 2022; Dendooven et al, 2023), leaving most of the CENP-C protein structure unresolved. This is surprising as CENP-C is a critical component of the CCAN (Klare et al, 2015; Walstein et al, 2021) and one of the kinetochore components that directly binds to CENP-A nucleosomes. Although the dimerization domain of CENP-C is essential for faithful chromosome segregation (Trazzi et al, 2009; Hara et al, 2023), the CCAN structures were reconstituted at a molar ratio of the CCAN components, including CENP-C fragments. Here, we observed that the dimensions of the smallest CENP-C complexes purified from HeLa cells (Figs 2 and 5) agree with the published CCAN in vitro structures. This leaves the composition and organization of the larger CENP-C complexes we purified to be determined. A logical factor in forming larger CENP-C complexes may lay in the CENP-C dimerization domain and its potential

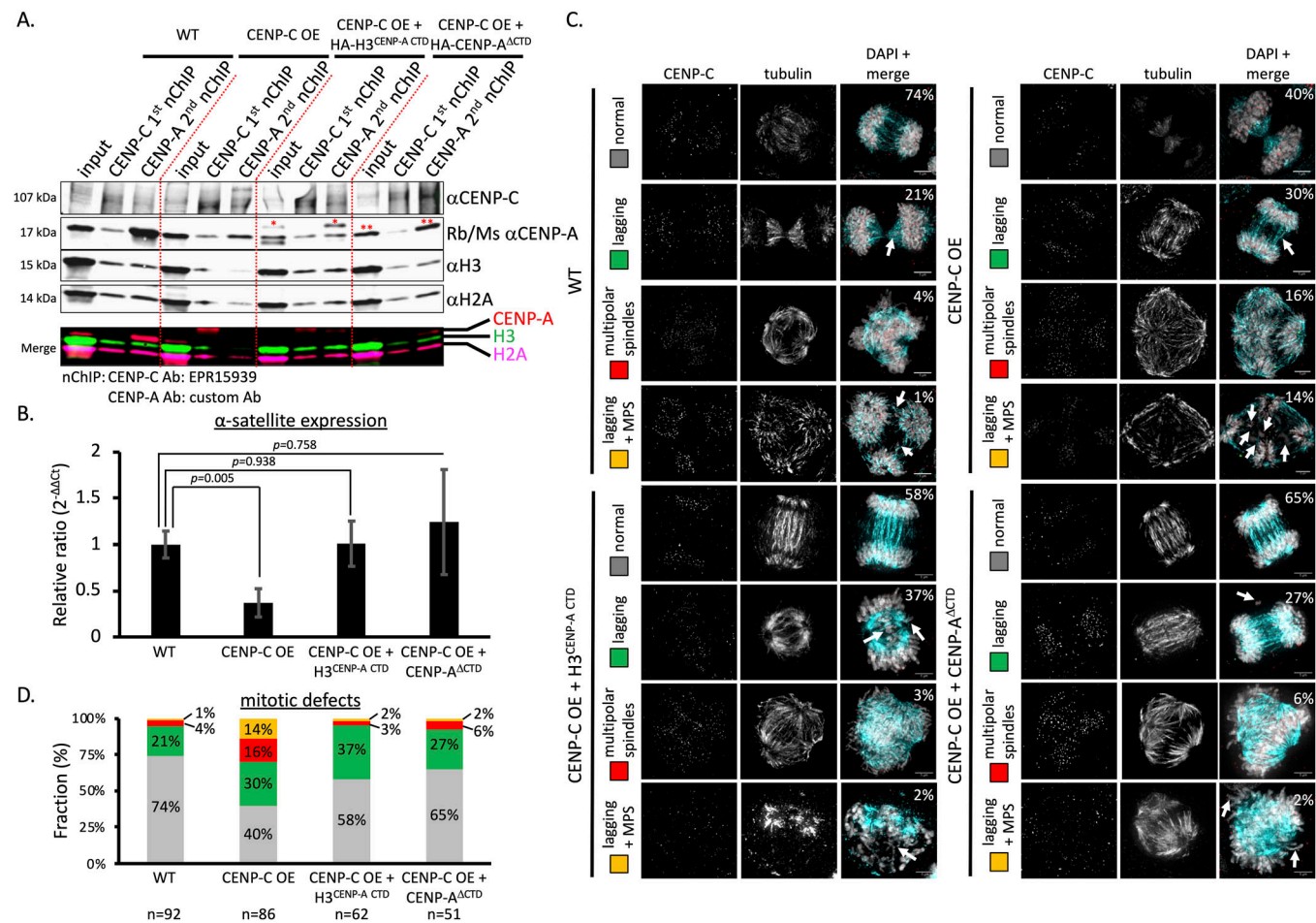

**Figure 7. CENP-A mutants partially rescue centromeric transcription and mitotic defects induced by CENP-C overexpression.**
**(A)** Western blot of chromatin extracted by a serial CENP-C nChIP followed by CENP-A nChIP of WT HeLa cells, HeLa cells where CENP-C is overexpressed (CENP-C OE), cells where CENP-C is overexpressed in the presence of H3$^{CENP-A\ CTD}$, and cells where CENP-C is overexpressed in the presence of CENP-A$^{\Delta CTD}$. A single red asterisk (*) = HA-H3$^{CENP-A\ CTD}$. A double red asterisk (**) = HA-CENP-A$^{\Delta CTD}$. **(B)** Relative expression (2$^{-\Delta\Delta Ct}$) of centromeric α-satellite DNA in all four conditions (error bars = SD; paired t test; significance was determined at $P < 0.05$). n = number of particles measured. **(C)** Representative images of all four conditions (WT, CENP-C OE, CENP-C OE + HA-H3$^{CENP-A\ CTD}$, and CENP-C + HA-CENP-A$^{\Delta CTD}$) for normal anaphase spreads, anaphase with lagging chromosomes (lagging), anaphase cells with multipolar spindles, and anaphase cells with lagging chromosomes and multipolar spindles (MPS). Lagging chromosomes are highlighted with white arrows. In the top right-hand corner of the DAPI + merge image is the percentage per category noted. CENP-C = red; tubulin = blue; DAPI = gray. n = number of cells quantified. Scale bar = 5 μm. **(D)** Cells in all four conditions were synchronized to the M phase to score mitotic defects. We observed that the level of both the multipolar spindle (red) and multipolar spindle with lagging chromosomes (orange) was reduced to WT levels. Source data are available for this figure.

to associate with up to four CENP-A nucleosomes per CENP-C dimer. Furthermore, it will be interesting to unravel what causes the increase in CENP-C complex height as cells progress through the cell cycle.

Surprisingly, we show that the large CENP-C complexes only associate with a fraction of the total CENP-A available in the nucleus. Indeed, using two different sets of CENP-C and CENP-A antibodies (Figs 1B and 4B), we pulled down excess CENP-A chromatin after we pulled down CENP-C and their associated CENP-A chromatin. The ectopic expression of CENP-A mutants did not alter this observation (Figs 6C and 7A). These results are in agreement with previous ChIP-seq and APEX-chromatin fiber FISH experiments (Henikoff et al, 2015; Kyriacou & Heun, 2018), where CENP-C levels were found to be lower than CENP-A levels. When we measured the dimensions of these two CENP-A populations, we found that the unbound CENP-A nucleosomes were shorter, in line with previous reports (Dalal et al, 2007; Dimitriadis et al, 2010; Bui et al, 2012, 2013; Athwal et al, 2015). In contrast, the CENP-C–associated CENP-A

nucleosomes were almost indistinguishable from bulk H3 nucleosomes (Figs 3C and 4C). The nucleosomes measured in this study were only the nucleosomes readily visualized by AFM. We cannot exclude that CENP-A nucleosomes were embedded in the CENP-C complex. This leaves the question what the CENP-A nucleosomes look like in the CCAN unanswered. Indeed, the position and structure of CENP-A nucleosomes at or within the CCAN also remains debated. Two groups that in vitro reconstituted the CCAN included an in vitro reconstituted CENP-A mononucleosome (Tian et al, 2022; Yatskevich et al, 2022), whereas another group tried to model a CENP-A nucleosome in the space within the CCAN (Pesenti et al, 2022). The structure of CENP-A nucleosomes in either situation was not the same. Our data paint a more complex picture in vivo with different CENP-A structures that can be found at the centromere. Nevertheless, it remains to be determined what underlies the structural flexibility of CENP-A nucleosomes in vivo or how these structural CENP-A variants can be reconstituted in vitro.

The overexpression of CENP-C results in compaction of CENP-A chromatin, reduced centromeric RNAP2 levels, inhibition of $\alpha$-satellite transcription, and impaired loading of new CENP-A molecules (Melters et al, 2019, 2023). Centromeric transcription and mitotic defects induced by CENP-C overexpression were partially rescued by HA-H3$^{\text{CENP-A CTD}}$ and HA-CENP-A$^{\Delta\text{CTD}}$ (Fig 7). Although lagging chromosome defects were not rescued by the CENP-A mutants, it is interesting that the multipolar spindle defects were rescued. Multipolar spindles are commonly formed because of centrosome defects (Lens et al, 2010). In fruit fly S2 cells, knockdown of the fruit fly CENP-A chaperone CAL1 resulted in multipolar spindles (Erhardt et al, 2008). Therefore, our results hint at a possible link between CENP-C:CENP-A chromatin and centrosomes. It will be an interesting avenue for future work to elucidate the potential interaction between centromeres and centrosomes in regulating multipolar spindles. Whether there is a link between centromeric transcription and centrosomes remains to be determined.

In summary, in this report we describe the isolation and description of endogenous CENP-C complex, whose dimensions are in line with published in vitro cryo-EM CCAN/kinetochore structures (Pesenti et al, 2018; Tian et al, 2022; Yatskevich et al, 2022) (Figs 1, 2, and 5) and these complexes associate with CENP-A nucleosomes with a distinct structure that indicates an altered resistance against the AFM tapping force (Figs 2 and 5). Altogether, the nucleosome binding protein CENP-C modulates centromeric homeostasis through modulation of centromeric chromatin dynamics by altering the transcriptional competency of CENP-A chromatin (Fig 7B). Nucleosome dynamics play an important role in genome compaction, protection from DNA damaging agents, and regulating DNA access by DNA binding factors. These dynamics are driven by only a few interactions between the interfaces of DNA and nucleosomes (Bowman & Poirier, 2015; Armeev et al, 2021; Peng et al, 2021). Previously, we described a CENP-A core post-translation modification that altered the binding of CENP-C to CENP-A in vivo (Bui et al, 2017), whereas CENP-C alters the material properties and mobility of CENP-A nucleosomes in vitro (Falk et al, 2015, 2016; Guo et al, 2017; Melters et al, 2019, 2023; Rakshit et al, 2020). An exciting line of ongoing investigation is to examine how DNA or histone modifications, or chromatin remodelers, which tilt the scale toward specific conformations of CENP-A nucleosomes, might alter its interactions with chaperones, kinetochore partners, and its occupancy on centromere $\alpha$-satellite DNA. It will also be of interest to investigate the altered primary and secondary structures of endogenous centromeric nucleosomes and whether nucleosome-stabilizing interactions promote or suppress centromeric transcription required for the epigenetic memory of centromeres in various species (Wong et al, 2007; Li et al, 2008; Chueh et al, 2009; Ferri et al, 2009; Bergmann et al, 2011; Choi et al, 2011; Ohkuni & Kitagawa, 2011; Chan & Wong, 2012; Quénet & Dalal, 2014; Rošić et al, 2014; Catania et al, 2015; Grenfell et al, 2016; McNulty et al, 2017; Bobkov et al, 2018; Zhu et al, 2018; Ling & Yuen, 2019; Melters et al, 2019).

# Materials and Methods

**Key resources table.**

| Reagents or resource | Source | Identifier | Application | Quantity |
|---|---|---|---|---|
| ACA serum | BBI Solutions | SG140-2 | nChIP | 5 µl |
| Anti-CENP-A (rabbit) | Custom-made | | nChIP | 3 µl |
| Anti-CENP-A (mouse) | Abcam | ab13939 | WB, IF | 1:1,000 |
| Anti-CENP-A (rabbit) | Abcam | ab45694 | WB | 1:3,000 |
| Anti-CENP-A (rabbit) | Millipore | 04-205 | WB | 1:3,000 |
| Anti-CENP-B | Active Motif | 61288 | WB | 1:500 |
| Anti-CENP-C (guinea pig) | MBL International | PD030 | nChIP, IF | 5 µl, 1:1,000 |
| Anti-CENP-C (rabbit) | Abcam | EPR15939 | nChIP, WB | 3 µl, 1:1,000 |
| Anti-CENP-C (rabbit) | Santa Cruz | sc-22789 | WB | 1:500 |
| Anti-CENP-N | Avivasysbio | ARP57258-P050 | WB | 1:500 |
| Anti-CENP-I | Bethyl Laboratories | A303-374A | WB | 1:1,000 |
| Anti-CENP-T | Bethyl Laboratories | A302-314A | WB | 1:1,000 |
| Anti-CENP-W | Invitrogen | PA5-34441 | WB | 1:300 |
| Anti-MIS12 | Abcam | ab70843 | WB | 1:500 |
| Anti-HJURP | Bethyl Laboratories | A302-822A | WB | 1:1,000 |
| Anti-H2A | Abcam | ab18255 | WB | 1:1,000 |
| Anti-H2B | Abcam | ab1790 | WB | 1:1,000 |
| Anti-H4 | Cell Signaling | 2935T | WB | 1:1,000 |
| Anti-H3 | Santa Cruz | sc-8654 | WB | 1:3,000 |
| Anti-HA | Santa Cruz | sc-805 | nChIP | 5 µl |
| Anti-GFP | Santa Cruz | sc-5385 | IF | 1:1,000 |
| Anti-mCherry | Invitrogen | M11217 | IF | 1:1,000 |

**Software and algorithms.**

| Software and algorithms | |
|---|---|
| RepBase | http://www.girinst.org/repbase |
| Gwyddion | http://gwyddion.net/ |
| R | https://www.r-project.org/ |
| NIH ImageJ | https://imagej.net/software/fiji |
| Bio-Formats | https://www.openmicroscopy.org/bio-formats/ |

### Transient transfection and cell lines

HA-tagged H3 with the CENP-A C-terminal domain (HA-H3[CENP-A CTD]), HA-tagged CENP-A without the C-terminal domain (HA-CENP-A[ΔCTD]), and mCherry-tagged CENP-A (mCherry-CENP-A) were driven by the CMV promoter plasmid and were made in-house and have been deposited to Addgene. GFP-tagged CENP-C was driven by the CMV promoter plasmid and was generously provided by Dan Foltz. Transfections were done using Neon Transfection System (Cat #MPK5000; Thermo Fisher Scientific) with 100 $\mu$l kit (Cat #MPK10096; Thermo Fisher Scientific), using the following parameters for HeLa cell lines: 2 pulses of 1050 V/30 ms. HeLa cells were acquired from Abcam (Cat #ab265233). HeLa were grown in DMEM (Cat #11965; Invitrogen/Thermo Fisher Scientific) supplemented with 10% FBS and 1X penicillin and streptomycin cocktail. Cells were harvested 72 h post-transfection for downstream applications.

### Native chromatin immunoprecipitation and western blotting

For cell cycle experiments, HeLa cells were synchronized by a double thymidine block (0.5 mM, Cat #T9250; Sigma-Aldrich) as described here (Melters et al, 2019). Cells were harvested in the S phase (3 h post-release), G2 phase (7 h post-release), M phase (10 h post-release), and G1 phase (15 h post-release). nChIP experiments were performed without fixation. After cells were grown to ~80% confluency, they were harvested as described previously (Bui et al, 2012, 2017). For best results for chromatin preparation for AFM, the pellet that is obtained after each spin-down during the nucleus extraction protocol (Walkiewicz et al, 2014a) is broken up with a single gentle tap. Nuclei were digested for 6 min with 0.25 U MNase/ml (Cat #N3755-500UN; Sigma-Aldrich) and supplemented with 1.5 mM CaCl$_2$. After quenching (10 mM EGTA), nucleus pellets were spun down, and chromatin was extracted gently, overnight in an end-over-end rotator, in low-salt solution (0.5X PBS; 0.1 mM EGTA; protease inhibitor cocktail (Cat #05056489001; Roche)). nChIP chromatin bound to Protein G Sepharose beads (Cat #17-0618-02; GE Healthcare) was gently washed twice with ice-cold 0.5X PBS and spun down for 1 min at 4°C at 60$g$. After the first nChIP, the unbound fraction was used for the sequential NChIP. Western blot analyses were done using LI-COR's Odyssey CLx scanner and Image Studio v2.0.

### Glycerol gradient sedimentation

A total of 2 ml of extracted chromatin was applied to 10 ml of 5–20% glycerol gradient containing 50 mM Tris–HCl, pH 8.0, 2 mM EDTA, 0.1% NP-40, 2 mM DTT, 0.15 M NaCl, and 1X protease inhibitor cocktail

layered over 0.4 ml of 50% glycerol. The chromatin was centrifuged with a SW41Ti rotor (Beckman) at 22,000 rpm in a Beckman Coulter Optima XPN-80 Ultracentrifuge for 15.5 h at 4°C. 1 ml aliquots were fractioned from the top, and DNA and protein samples were separated by either 1.2% agarose gel electrophoreses or 4–20% SDS–PAGE gels, respectively. Serial NChIP was performed on all 12 fractions.

### AFM and image analysis

Imaging of CENP-C and CENP-A nChIP and bulk chromatin was performed as described previously (Dimitriadis et al, 2010; Walkiewicz et al, 2014a) with the following modifications. Imaging was performed using standard AFM equipment (Oxford Instruments, Asylum Research's Cypher S AFM, Santa Barbara, CA) with silicon cantilevers (OTESPA or OTESPA-R3 with nominal resonances of ~300 kHz, stiffness of ~42 N/m, and tip radii of 3–7 nm) in non-contact tapping mode. 10 $\mu$l of bulk, CENP-A, or CENP-C chromatin sample was deposited on APS-treated mica (Dimitriadis et al, 2010; Walkiewicz et al, 2014a). The samples were incubated for 10 min, rinsed gently to remove salts, and dried mildly under vacuum before imaging. Automated image analysis was performed as described in Walkiewicz et al (2014a) with the only modifications that R software was used instead of Microsoft Excel. A total of six biological replicates were performed for CENP-C experiments and three biological replicates for both the CENP-A and bulk chromatin experiments. Bulk chromatin from the same preparation was imaged in parallel to get the baseline octameric range. For all samples, manual spot analyses were performed to confirm the accuracy of automated analyses.

### Immuno-AFM

In vitro reconstitution of CENP-A (CENP-A/H4 Cat #16-010, and H2A/H2B Cat #15-0311, EpiCypher, Research Triangle Park, NC) and H3 (H3/H4 Cat #16-0008, and H2A/H2B Cat #15-0311, EpiCypher Research Triangle Park, NC) nucleosomes was performed as previously described (Dimitriadis et al, 2010; Walkiewicz et al, 2014a). Chromatin from HeLa cells was obtained from fractions 6 and 7 of a glycerol density gradient (containing on average tri-, tetra-, and penta-nucleosome arrays). These samples were subjected to immuno-AFM as described previously (Browning-Kelley et al, 1997; Cheung & Walker, 2008; Banerjee et al, 2012). An aliquot of each sample was imaged by AFM in non-contact tapping mode. The remainder of the samples were incubated overnight at 4°C with anti-CENP-A antibody (Cat #ab13939; Abcam) in an end-over-end rotator before being imaged by AFM. Finally, these samples were incubated with anti-mouse secondary antibody (LI-COR's IRDye 800CW Donkey anti-Mouse IgG, Cat #925-32212) for an hour at room temperature in an end-over-end rotator and imaged by AFM in non-contact tapping mode. We analyzed the height profiles of the nucleosomes and antibody complexes as described above.

### TEM

For TEM, the NChIP samples were fixed by adding 0.1% glutaraldehyde at 4°C for 5 h, followed by 12-h dialysis against HNE buffer (10 mM Hepes, pH 7.0, 5 mM NaCl, 0.1 mM EDTA) in 20,000 MWCO

membranes dialysis cassettes (Slide-A-Lyzer Dialysis Cassette, Cat #66005; Thermo Fisher Scientific) at 4°C. The dialyzed samples were diluted to about 1 $\mu$g/ml concentration with 67.5 mM NaCl, applied to carbon-coated and glow-discharged EM grids (T1000-Cu; Electron Microscopy Sciences), and stained with 0.04% uranyl acetate. Dark-field EM imaging was conducted at 120 kV using a JEM-1000 electron microscope (JEOL USA) with a SC1000 ORIUS 11 megapixel CCD camera (Gatan, Inc.).

### Immunostaining of interphase nuclei and mitotic chromosomes

For mitosis analysis, HeLa cells were synchronized to mitosis with a double thymidine block and subsequently deposited onto a slide by cytospin. Cells were prefixed with 1% paraformaldehyde in PEM (80 mM K-PIPES [pH 6.8], 5 mM EGTA [pH 7.0], 2 mM MgCl$_2$) for 10 min at RT. After three washes with ice-cold PEM, soluble proteins were extracted with 0.5% Triton X in CSK (10 mM K-PIPES [pH 6.8], 100 mM NaCl, 300 mM sucrose, 3 mM MgCl$_2$, 1 mM EGTA) for 5 min at 4°C. Slides were rinsed with PEM, and a few drops at a time of 4% paraformaldehyde were dribbled on the slides and incubated for 20 min at 4°C. After three washes with PEM, cells were permeabilized with 0.5% Triton X in PEM for 5 min at RT, followed by three washes with PEM. Slides were incubated with blocking solution (PBS + 3% BSA + 10 $\mu$l/ml normal goat IgG) for 1 h at RT. Primary antibodies (CENP-C, GFP, mCherry) were used at dilution 1:1,000 in hybridization buffer (1x PBS + 1% BSA) and incubated overnight at 4°C. After three washes with 1x PBS + 0.1% Tween-20, Alexa secondary antibody (488, 647; dilution of 1:1,000 in hybridization buffer) was added and incubated for 1 h at RT, followed by three washes with 1x PBS + 0.1% Tween-20 and two washes with 1x PBS. Slides were air-dried for 30 min at RT, coverslips were mounted with VECTASHIELD with DAPI, and coverslip edges were sealed with nail polish. Images were obtained using the DeltaVision RT system fitted with a CoolSNAP charge-coupled device camera and mounted on Olympus IX70. Deconvolved IF images were processed using ImageJ. To determine colocalization between the two channels, the ImageJ build-in feature for colocalization was used. Mitotic defects (lagging chromosomes and/or multipolar spindles) were counted for 92, 86, 62, and 51 cells (mock, GFP-CENP-C, GFP-CENP-C + H3$^{CENP-A\ CTD}$, and GFP-CENP-C + CENP-A$^{\Delta CTD}$, respectively).

### ChIP-seq

CENP-C NChIP followed by ACA NChIP was conducted, as well as an IgG NChIP and input control as described above. Next, DNA was isolated after treating with proteinase K the samples, followed by DNA extraction by phenol–chloroform. The samples were used to prepare libraries for PacBio single-molecule sequencing as described in the manufacturer's protocol (PacBio). Libraries were produced and loaded on a ZMW chip either by diffusion or after size selection of the inserts (>1,000 bp) for all four samples. Subsequently, the reads were sequenced on PacBio RS II operated by Advanced Technology Center, NCI. Read depths obtained for each sample were ACA-ChIP (23,785), CENP-C ChIP (6,447), IgG (121,874), and input (88,992). GC contents ranged from 42.4% to 44%. Reads were confirmed to be free of residual PacBio internal control

sequence via alignment with Bowtie2 (v.2.3.4) with default parameters (Langmead & Salzberg, 2012). Average raw read lengths for each sample were ACA-ChIP (1,624 nt), CENP-C ChIP (1,920 nt), IgG (1,236 nt), and input (1,344 nt). Sequence reads were mapped to either sequence in RepBase (ALR, ALR1, and ALR2), the consensus sequence used by Hasson et al (2013), and the consensus sequences used by Henikoff et al (2015), using Bowtie2 in sensitive–local mode.

### Quantitative PCR

$\alpha$-Satellite expression levels in HeLa cells that were either mock-transfected or transfected GFP-CENP-C were detected as previously described (Quénet & Dalal, 2014; McNulty et al, 2017). RNA was extracted and quantified by UV spectroscopy, and equal quantities were retrotranscribed using the Superscript III First-Strand Synthesis kit as described above. Complementary DNA (cDNA) samples were prepared using the iQ SYBR Green supermix (#170–8880; Bio-Rad) following the manufacturer's protocol. Control reactions without the cDNA were performed to rule out non-specific amplification. The qPCR was run on StepOnePlus Real-Time PCR System (Applied Biosystems). Primer sequences are as follows.

**Primer sequences used for qPCR.**

| Oligonucleotides | Sequence |
|---|---|
| Centromeric $\alpha$-satellite Forward | 5'-CATCACAAAGAAGTTTCTGAGAATGCTTC-3' |
| Centromeric $\alpha$-satellite Reverse | 5'-TGCATTCAACTCACAGAGTTGAACCTTCC-3' |
| GAPDH Forward | 5'-GCGGTTCCGCACATCCCGGTAT-3' |
| GAPDH Reverse | 5'-CCCCACGTCGCAGCTTGCCTA-3' |

The comparative cycle threshold (C$_T$) method was used to analyze the expression level of $\alpha$-satellite transcripts. C$_T$ values were normalized against the average C$_T$ value of the housekeeping gene GAPDH. Relative fold differences ($2^{-\Delta\Delta CT}$) are indicated in Fig 5C.

### Quantification and statistical analyses

Significant differences for Western blot quantification and nucleosome height measurements from AFM analyses were performed using either paired or two-sided $t$ test or one-way ANOVA followed by Tukey's HSD multiple comparisons of the mean as described in the Fig legends. Significance was determined at $P < 0.05$.

## Data Availability

The ChIP-seq sequence data and additional experimental detail can be found in the Gene Expression Omnibus (GOE) under the accession number GSE129351.

# Supplementary Information

# Acknowledgements

We thank Drs. Tom Misteli and Sam John, and members of the CSEM laboratory for critical comments and suggestions. We thank Dr. Kerry Bloom for encouraging us to purify kinetochore-bound centromeric chromatin. We acknowledge Dr. Andrea Musacchio's suggestion to test for the function of the unbound CENP-A population. We thank the reviewers for pushing us to do the cell cycle characterization of the CENP-C complex. This work used the computational resources of the NIH HPC Biowulf cluster (http://hpc.nih.gov). DP Melters, M Bui, T Rakshit, D Sturgill, and Y Dalal. were supported by the Intramural Research Program of the Center for Cancer Research at the National Cancer Institute/NIH. SA Grigoryev was supported by NSF grant 1911940.

## Author Contributions

DP Melters: conceptualization, data curation, formal analysis, investigation, visualization, methodology, and writing—original draft, review, and editing.
M Bui: investigation, methodology, and writing—review and editing.
T Rakshit: investigation, methodology, and writing—review and editing.
SA Grigoryev: investigation, methodology, and writing—review and editing.
D Sturgill: software and investigation.
Y Dalal: conceptualization, supervision, project administration, and writing—original draft, review, and editing.

## Conflict of Interest Statement

The authors declare that they have no conflict of interest.

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
