## [Reviewer comments · Life Science Alliance]

Life Science Alliance

High-Resolution Analysis of Human Centromeric Chromatin

Daniël P Melters, Minh Bui, Tatini Rakshit, Sergei A Grigoryev, David Sturgill and Yamini P Dalal

DOI: <https://doi.org/10.26508/lsa.202402819>

Corresponding author(s): Dr. Yamini P Dalal (National Institutes of Health)

Review Timeline:

Submission Date:	2024-05-13
Editorial Decision:	2024-06-10
Revision Received:	2024-12-04
Editorial Decision:	2024-12-23
Revision Received:	2025-01-10
Editorial Decision:	2025-01-13
Revision Received:	2025-01-15
Accepted:	2025-01-16

Transaction Report:

June 10, 2024

Re: Life Science Alliance manuscript #LSA-2024-02819

Dr. Yamini P Dalal
National Institutes of Health
National Cancer Inst. (NCI)
41 Library Drive MSC 5055
Bethesda, MD 20892-5055

Dear Dr. Dalal,

Thank you for submitting your manuscript entitled "High-Resolution Analysis of Human Centromeric Chromatin". The manuscript has been evaluated by expert reviewers, whose reports are appended below. Unfortunately, after an assessment of the reviewer feedback, our editorial decision is against publication in Life Science Alliance.

Although your manuscript is intriguing, I feel that the points raised by the reviewers are more substantial than can be addressed in a typical revision period. If you wish to expedite publication of the current data, it may be best to pursue publication at another journal.

Given the interest in the topic, I would be open to re-submission to Life Science Alliance of a significantly revised and extended manuscript that fully addresses the reviewers' concerns and is subject to further peer review. If you would like to resubmit this work to Life Science Alliance, you may submit an appeal directly through our manuscript submission system. Please note that priority and novelty would be reassessed at re-submission.

Regardless of how you choose to proceed, we hope that the comments below will prove constructive as your work progresses.

Thank you for thinking of Life Science Alliance as an appropriate place to publish your work.

Sincerely,

Reviewer #1 (Comments to the Authors (Required)):

The manuscript #LSA-2024-02819 aims at characterizing centromere-specific chromatin using nanoscale imaging techniques and biochemical approaches to better document the hierarchy of protein complexes, among which are the kinetochore and the constitutive centromere associated network (CCAN), assembled at centromeric loci.

At centromeres, it has been well-documented that CENP-A provides an epigenetic and structural foundation for the recruitment of inner kinetochore proteins, among which is CENP-B, which in turn recruits the CCAN. The project that forms the core of this manuscript stemmed from the observation by others of a large and compact macromolecular complex identified through immunoprecipitation of native CENP-A chromatin (nChIP).

The authors then decided to tackle the question of the identity and structure of this never-studied macro-complex.

Here, they revealed that the CENP-C complex is a large polygonal structure that is physically associated with four to six CENP-A nucleosome and contains CCAN members. Surprisingly, this complex associates with only a small fraction of the total nuclear CENP-A with a large fraction of CENP-A nucleosomes not being associated with CENP-C. Their data suggest that this large complex represents the inner kinetochore/CCAN with nucleosome-free DNA organized by CENP-C. They also attempted to link a CENP-C overexpression context to increased centromeric transcription and mitotic defects, and its rescue by CENP-A mutants, although this part and the rationale for it are the least convincing.

As a whole, the manuscript is interesting for the centromere community and is based on rather solid and convincing imaging at the nanoscale. The Development of the immuno-AFM approach is also new and very interesting. However, I found the whole work difficult to follow. Compared to their current MS in BioRxiv, part of which has been published in eLife 2023, the present MS

is far less well written and described, suffers from expeditive writing and conclusions and is missing a clear rationale for studies in several places. The redundancy with their previous manuscripts (Melters 2019 and 2023) is also somewhat disturbing. To finish, I am afraid that the claim that the data support the model of "a balance between unbound and kinetochore bound CENP-A chromatin is needed for supporting a unique higher-order structure at centromeric chromatin" remains rather weak at this stage. Hence, I would suggest major modifications throughout the text.

The manuscript should be improved in terms of writing, especially for the description of the approaches used in the results section so that the reader does not need to search the information in the methods section and in the legends to figures.

The abstract does not seem to really match conclusions drawn from the work:

- CENP-C complex-bound chromatin has strongly reduced DNA accessibility?
- the CENP-C complex matches the dimensions of both a single and multiple CCAN?

There are many places that need further clarification - Examples follow:

line 102, "therefore": what is the link between "small fraction of macromolecular complexes within CENP-A native ChIP » and developing an assay to purify CENP-C-bound chromatin?

Paragraph "CENP-C complexes are diverse in size":

- an unclear title since the paragraph deals with the large polygonal complexes that came down with CENP-C nChIP. What characterizes the different complexes at this point?
- Please precise what definition you used for the Feret diameter. The whole calculation is a little bit convoluted. How a surface with 15-17 nm sides can make a circle of 22 nm? Why it "would have" a height of 9.3 nm?
- Apparent contradiction between "CENP-C complexes were associated with... nucleosome-free DNA" (line 117) and "CENP-C can... potentially associate with up to four CENP-A nucleosomes" (line 142) and also "CENP-A nucleosomes are indeed associated with the purified CENP-C complexes" (line 182).
- What is a stepwise correlation in this context? (line 143 and line 146-147).

Line 217, "there is a ~6-fold excess of CENP-A that is not strongly associated with CENP-C": precise if nucleosomal CENP-A?

Methods: PacBio sequencing would deserve more information: how many reads, length of fragments, QC, alignment, which "sequences in Repbase"? etc...

What is the logic of confronting "a critical balance must be maintained between kinetochore-bound CENP-A chromatin and kinetochore-free CENP-A chromatin" with "a CENP-C overexpression scenario"?

Lines 285-287: predictions should be experimentally tested and experimentally validated

Lines 289-290: Explain the rationale for looking at centromeric transcription

Line 291, "a ~60% reduction in a-satellite transcription": what are the basic levels in HeLa cells? Please comment on the fact that centromeres are supposed to be repressed in "normal" cells (control set at 1, so is not informative). This sentence is also found in their previous paper, so please quote your work when it is redundant with previous papers.

Please provide controls for H3CpA CTD or CENP-A CTD ectopic expression. What is their localization in transfected cells or mitotic chromosomes? Please provide controls for "CENP-A mutants that function either as a sink or are unable to bind CENP-C".

Line 301: to what relate (40% normal, 60% abnormal) and (74% normal, 36% abnormal).

Line 306: the "rescue" of mitotic defects is not convincing as there is no statistical analysis and lagging chromosomes remain virtually unchanged. This should be discussed.

Lines 309-315, this concluding paragraph raises some questions:

- What happens to endogenous CENP-C upon CENP-C ectopic expression? Same for CENP-A. What is the localization of mutants CENP-A?
- In these conditions, the authors should assess the impact on kinetochore-bound and kinetochore-unbound CENP-A chromatin

The discussion should be revised in the light of previous comments.

Figure 3 is puzzling at first sight because the panel in column 2/raw 2 is not at the same scale as the other images.

Reviewer #2 (Comments to the Authors (Required)):

General summary & opinion:

In the present study, Melters et al. analyze the structure of chromatin-bound CENP-C in human cells. For this, using the method described in their previous studies (Dimitriadis et al 2010, Walkiewicz et al 2014), they performed native chromatin-immunoprecipitation (nChIP) of CENP-C in HeLa cells. They further pursue their analysis using single molecule imaging (as described in their publication Melters & Dalal 2021) and biochemical approaches. They found that CENP-C associated with a large complex bound to CENP-A, a complex more resistant to nuclease digestion than bulk or CENP-A containing chromatin. This complex could correspond to a kinetochore-bound complex associated with CENP-A, with reduced DNA accessibility. Using serial nChIP with first CENP-C followed by ACA antibodies, the authors also revealed two possible CENP-A nucleosomal configurations: (i) a small portion associated with CENP-C and (ii) the vast majority remaining, not associated with CENP-C. Finally, authors performed CENP-C overexpression experiments. This overexpression led to 60% reduction in alpha-satellite transcription, similarly to previous result (Melters et al 2023), along with mitotic defects. These defects were rescued by the introduction of CENP-A mutants. Overall, the authors conclude that based on CENP-C association, two populations of CENP-A-nucleosomes co-exist in human cells. They further propose that a proper balance between these populations would be crucial for proper centromere function:

- The CENP-C unbound population would maintain centromeric transcription and CENP-A levels at centromeres,
- The CENP-C bound population (along with kinetochore proteins) would ensure faithful chromosome segregation.

CENP-A nucleosomes display specific features including looser DNA binding and increased dynamics of DNA ends (Sekulic et al 2010, Panchenko et al 2011, Tachiwana et al 2011, Lacoste et al, 2014, Ali-Ahmad et al 2019; Arimura et al 2019). But how this local flexibility and CENP-A interactors together impact higher-order structure of centromeric chromatin remained unclear. These specific features might lead to a dynamic structure and may also involve compact structures (Ali-Ahmad et al 2019, Melters et al 2019, 2023). Importantly, within the CCAN how CENP-A nucleosomes along with their distinct structures are placed is still enigmatic. In this context, this manuscript, which represents a follow-up of previous studies from the same authors (Melters et al 2019, 2023) is relevant. It confirms the existence of two types of CENP-A nucleosomal complexes as hypothesized in previous studies (Henikoff et al 2015, Kyriacou & Heun 2018). Importantly, to document the importance/role of these two types of structures would represent a valuable contribution for the field. However, the second part of the study deserves to be revisited to firm up the conclusions. Importantly, series of concerns remains as listed below in detail, along with suggestions to help to address them and better substantiate the conclusions. These points should be considered carefully before publication.

Major concerns:

(1) A main concern relates to their final model which argues that the balance between the two CENP-A fractions is crucial for centromere function. To draw this conclusion, additional experiments should be performed:

- First, to confirm that the overexpressed CENP-C actually binds CENP-A chromatin and change the balance between CENP-A free and bound-CENP-A nucleosomes (saturates CENP-A nucleosomes).
- Second, to provide data on whether the rescue with CENP-A mutants does restore a proper balance between CENP-C-free and -bound nucleosomes.

(2) The serial nChIP (CENP-C & ACA) in asynchronous cells revealed two CENP-A nucleosomal configurations is an interesting observation. The authors could comment on the choice of ACA serum, instead of CENP-A antibody. To strengthen this finding, a CENP-A nChIP should reveal these two fractions, bound or not by CENP-C.

- If the authors had already performed this experiment in a previous study, they should make sure that the information is made available (referring to precise figures/experiments in the paper). Otherwise, this should be documented in this manuscript.
- Possibly, an approach with synchronized cells would enable to discard possible differences reflecting possible cell cycle variations such as cell cycle coupled centromeric chromatin remodelers, or ectopic CENP-A assembled onto chromosome arms in early G1 (Nechemia-Arbely et al 2019).

(3) The authors predict that CENP-A mutants function as sink for excess CENP-C (H3CpA-CTD) or increase the kinetochore-free CENP-A population (CENP-A Δ CTD), page 13. To support the following conclusions, the study should confirm that H3CpA-CTD mutants bind CENP-C, and that CENP-A Δ CTD mutants bind centromeric chromatin. Eventually, an IF analysis of CENP-A and centromeric markers in CSK extracted cells should provide more information on the effect of the CENP-A mutants.

Series of controls or specifications are also missing:

(1) In figure 5D, important markers should be added to control the ChIP-Western analysis:

- The HJURP staining should be added.
- The CENP-B staining, mentioned in the text (line 236), is missing in the figure and should be added too as it is an important control (as a CCAN member and one of the targets of the ACA antibody used).

(2) In figure 1C, the legend should indicate for the 3D rendering to which conditions it does correspond to. In addition, the figure should show 3D rendering images for both CENP-A and CENP-C pull downs (with the height scale) for comparison, and not only one condition.

(3) The manuscript should indicate the levels of CENP-C overexpression, upon transfection, compared to normal expression

levels. This is important to discuss the issue of balance. When does the tipping over occur ? Is there a threshold ?

(4) Same comment about the CENP-A levels upon addition of CENP-A mutant constructs. To be able to compare the results, it is important to ensure that comparable levels of expression have been achieved.

- Legend of suppl. Table S4 mentions the quantification "from Western blot" of CENP-A levels in the nChIP fractions but the result is not shown. This data should be added to the manuscript, along with the total CENP-A levels in whole cell extracts in each condition.

- In parallel, for establishing their model, the authors should consider the impact of high CENP-A levels on its localization (Lacoste et al 2014), and the fact that ectopic CENP-A can recruit CENP-C at ectopic regions (Van Hooser et al., 2001; Gascoigne et al., 2011; Lacoste et al., 2014).

(5) The CENP-A mutant constructs should be added to the manuscript. The experimental protocol to introduce the "CENP-A mutants" in cells is actually missing in the material & methods section of the manuscript and should be provided.

(6) Figure 8A should indicate what is revealed (the antibodies/colours used). The scale bar values should be added, at least in the legend. The figure should also mention the condition in which condition the images were taken. In addition, there should be representative images for each of the 4 conditions (WT/Mock, CENP-C OE, the 2 rescue experiments).

(7) ChIP & FISH data have shown that centromeric CENP-C levels are lower compared to those of CENP-A (Henikoff et al 2015, Kyriacou & Heun 2018). The mapping of CENP-C and CENP-A on APEX-chromatin fibers in Kyriacou & Heun 2018 showed that CENP-C is organized differently from CENP-A (a fraction of CENP-C localizes at places where CENP-A is absent) inside the centromeric domain. This is in line with the first conclusion of the study, and the authors should mention/discuss these previous reports.

Minor corrections & additional comments or questions:

(1) Could you detect micronuclei, following cell division? And do the cells keep cycling/proliferating as in the control condition?

(2) Figures 5/4/3/7: Please indicate in the legend what the error bars correspond to (even when it appears in the table of the supplementary data).

(3) In figure 2D, what does each point of the scatter plot represent? The legend should indicate if each dot represents a single or averaged measure...

(4) Figure 4 should be refined. In addition, it could possibly be part of figure 3...

- The plots seem to represent the height of CENP-A nucleosomes measured by AFM in the two fractions (CENP-C nChIP followed by CENP-A nChIP). The legend should be reformulated to indicate all the information clearly (height of CENP-A nucleosomes measured by AFM, in the CENP-C nChIP or in the CENP-A nChIP that follows, with the same anti-CENP-A & secondary antibodies as in figure 3).

- The legend mentions 4 conditions whose measures would be quantified in the figure: Bulk chromatin, CENP-A nucleosomes pulled down by CENP-A NChIP, CENP-A nucleosomes pulled down by CENP-C nChIP and "in addition ... large structures that came down by CENP-C NChIP". There are only 3 box plots and thus, it is unclear what condition is measured for each box plot.

- The figure should show representative images for each condition measured.

(5) Figure 5: The authors could add the molecular weights on the Western blot analysis.

(6) Figure 7: The name of the legend is incorrect: It is the same as in figure 8.

(7) In figure 8B, it is not mentioned what the "n" counts refer to precisely. In addition, quantification of mitotic defects should be represented more precisely, with bar plots indicating error bars for the quantification.

(8) Line 25: Sources should be indicated for the following sentence: "Inaccuracies in kinetochore assembly can lead to the formation of dicentric chromosomes, or chromosomes lacking kinetochores. In either case, chromosomes fail to segregate faithfully, which drives genomic instability."

(9) Line 31-32: their statement: "At the base of the inner kinetochore is the histone H3 variant CENP-A (CENH3), which provides the epigenetic foundation to assemble the kinetochore, recruiting inner kinetochore proteins CENP-B, CENP-C and CENP-N." relate here to a human system, thus it would be important to specify the following points.

- The sources Régnier et al and Mendiburo et al are indicated, the model used (chicken, Drosophila) for each publication should be specified.

- In addition, since this study relies on human cells as a model, sources for human cells could be added (Carroll et al 2009, 2010, Allu et al 2019).

(10) Line 109: "we observed large polygonal structures, with a roughly circular footprint". The authors could indicate the % of appearance of this type of structure. In other terms, did this pattern (shown in figure 1 and in representative images of figure S1) appear in 100% cases (177 out of the 177 measures) or at another, very high frequency?

(11) Line 150: "The precise composition of individual native CENP-C complexes is unknown, but mass spec analysis of tagged CENP-A nucleosomes brought down both kinetochore and non-kinetochore proteins (Foltz et al, 2009; Roulland et al, 2016; Remnant et al, 2019)." The reference Dunleavy et al 2009 should be added since it showed by immunostaining how, upon HJURP downregulation (and subsequent loss of centromeric CENP-A), CENP-C association with centromeres is decreased by contrast with CENP-B, for example.

(12) Line 241: The "Purified CENP-C complex robustly represents the CCAN components". To a lesser extent, this seems also true for the ACA-targeted (CENP-C-free) complex. The corresponding figure (5D) should be completed with the quantification of the Western blot, and the sentence should be phrased with a little caution (e.g., something like "CCAN components are more enriched in the purified CENP-C complex than the CENP-C-free complex, indicating that...")

(13) Line 359: "In many solid tumors, CENP-A is overexpressed and found outside the centromere by high-jacking H3.3 chaperones (Lacoste et al, 2014; Athwal et al, 2015; Zhao et al, 2016; Shrestha et al, 2017, 2021; Nye et al, 2018; Arunkumar et

al, 2022) or by altering the H3-H4 histone supply (Balachandra et al, 2024)." Although CENP-A is frequently overexpressed in cancer, its mislocalization has actually only been documented in cell lines. To be accurate, the sentence should be rephrased to highlight this important link between CENP-A overexpression in tumors and cell lines and CENP-A mislocalization in cell lines, along with the work of Verrelle et al. (2021), which gives hints in terms of changes in CENP-A subnuclear distribution in normal and tumoral tissues.

(14) Line 366: "CENP-A is preferentially found at genomic sites of high histone turnover..." The reference Lacoste et al 2014 could be added.

(15) Line 368: We suggest rephrasing the sentence for clarity purposes, for example: "The two following questions ensue: first, what is the mechanism by which lncRNA recruit new CENP-A nucleosomes, and second, what is the role of CENP-C in this process?"

(16) The section "Material and Methods" mentions the CENP-A antibody used for NChIP figure 1 was "custom made". If possible, the authors should provide a reference or an immunostaining and data further confirming the specificity of the antibody.

(17) The section "Material and Methods" describes protocols for a double thymidine block & for a quench-pulse-chase experiment. These experiments do not appear in the main manuscript. The experimental procedures that were not used in the study should be removed.

(18) Figure 2B should indicate from what experiment the image comes from (xChIP, nChIP...)

Reviewer #3 (Comments to the Authors (Required)):

This manuscript by Melters et al entitled seeks to answer an important and long-standing question regarding how centromeric chromatin conforms when bound to CENP-C, one of its direct binding partners. Here, they purify centromeric chromatin and view these purified constructs by AFM and TEM. They demonstrate some beautiful AFM of CENP-A nucleosomes bound to CENP-C. They identify both CENP-C bound and unbound CENP-A conformations, with CENP-C bound chromatin arranging into clusters of nucleosomes around a large polygonal (CCAN) structure, generally refractory to MNase digestion. The authors previously showed that CENP-C overexpression reduces transcription (Melters et al, 2023), and they correlate this with general inaccessibility of CENP-C-bound chromatin. However overall, I found this manuscript rushed, lacks attention to illustration and reader viewability, and would need significant improvement for my approval at this stage.

Major Points:

1. Figure 1 provides one of the major advances in this manuscript - an ability to image CENP-C bound centromeric chromatin. This is an excellent achievement and I commend the authors here. This figure would substantially be improved by a schematic illustration of the experiment from the outset. It takes a lot of cross referencing with legends and text to figure out what each figure represents. Point features out and don't assume the reader knows what they are looking at. This figure needs to be viewed in tandem with Fig S1 so I suggest the same there. Please keep this in mind where relevant for other figures also.

2. Figure 4 contains a measurement of nucleosome height. Can the authors include images of and point out these differing nucleosome heights? Some clear images of exactly what is being measured here, with plenty of examples, are absolutely required.

3. There is a strikingly large subfraction of CENP-A not associated with CENP-C. However, I worry to what extent this is over estimated, or simply just non-centromeric CENP-A. FRAP studies have shown CENP-C to be quite dynamic throughout the cell cycle, with stability during S-phase and mitosis (Hemmerich et al 2008, JCB). In addition, the Fukagawa lab have shown that dynamic changes in CCAN organization during progression of the cell cycle (in DT40 cells). I worry many of these effects we see associated with CENP-C are a consequence of the cell cycle stage and overall dynamics - and only the stable CENP-A/CENP-C interactions are being picked up. The diversity in nucleosome size in Figure 2 probably supports this.

All of these experiments have been done using asynchronous HeLa cells. I feel this manuscript would be significantly more informative with a cell cycle characterisation of this centromeric chromatin. This is likely to change throughout the cell cycle up until "final" point through mitosis. How these CENP-C-bound CENP-A chromatin look under AFM/TEM at different points of the cell cycle is worth looking at and might clear up some variability. In addition, we already know that CENP-A gets seeded ectopically on chromosome arms at some frequency and is not kinetochore bound. Other very recent publications from the lab make reference to this (e.g. Bui et al, 2024). Nechemia-Arbely et al, (Nat Cell Bio, 2019) suggest an error correction mechanism during S-phase whereby CENP-C protects centromeric CENP-A through replication, and ectopic CENP-A gets removed. If the vast majority of your cells are in G1, this may also be the reason you have a large majority of unbound CENP-A. This should be cleared up.

5. Figure 5 contains a sequential NChIP assay. However, ACA does not strike me as the appropriate antibody for the second NChIP in this case. Using a serum that contains CENP-A, CENP-B and CENP-C together is possibly not the cleanest approach. Can the authors explain their reasoning here? The fact that CENP-C complex associates with the CCAN kinetochore supports what this circular structure in the AFM images might be, and I agree it probably is based on Fig 5D. In line 237/238, the authors say that HJURP was enriched in the CENP-C NChIP. No data in main or supplemental figures show this, nor is it referenced in

the text that I can see. Please include this data.

6. The idea of expressing CENP-A mutants that can either sequester away excess CENP-C or which are insensitive to CENP-C overexpression is an interesting and entirely plausible hypothesis. The authors previously showed that CENP-C overexpression significantly reduced alpha-satellite transcription (Melters et al, *Elife*, 2023). The qPCR data in Fig 7B restates this and includes their mutant lines that restore expression levels. This is generally well-executed. How can the authors reconcile the misregulation of centromere transcription in many cancers, with the known fact that CENP-C is not overexpressed in solid tumours (per their discussion)? Overall I see Figure 7 as an opportunity missed without further exploration. After going to the trouble of making these mutants, why not put these constructs into their AFM/TEM assays? This may give you a more information on how CENP-C influences chromatin organisation. The H3CpA CTD mutant would be particularly interesting, and worth including.

8. In Figure 8, the primary mitotic defect (lagging chromosomes), has not been rescued. The multipolar spindle defects are the predominant rescue (as the authors mention). Can the authors explain this and why they think this is important? Why would multipolar spindles be the only phenotype that this would rescue? As things stand, I don't find this figure particularly informative.

Minor Points:

1. Is it always only one large structure associated with CENP-A chromatin?
2. Does bulk chromatin differ substantially from CENP-A NChIP chromatin? It looks so based on TEM images in Fig 1 & S1.
3. Figure legend title error: Figure 7 and Figure 8 have the same title.
4. Fig 7, 8 and possibly 9 are quite descriptive and can be combined into one figure depending on revisions.
5. In the discussion (which is in general overly long), the authors discuss the potential of lncRNAs at the centromere. CCTT is another lncRNA that has been shown to directly bind and recruit CENP-C to the centromere (Zhang et al, *Mol Cell*). It makes sense that if CENP-C is so flexible, and binds multiple CENP-A nucleosomes, that a lncRNA might help stabilise this CENP-C interaction. This is worth citing in their discussion.

We thank the reviewers for their insightful and thoughtful comments. We have addressed all major concerns raised, which has led us to reinterpret our results. Our first interpretation of the CENP-A mutant rescuing centromeric transcription and partially rescuing mitotic defects was by restoring the balance between the two CENP-A populations we identified. Results from additional experiment indicate a more nuanced picture, which we discuss in detail below. In short, we have extended our work by adding a comprehensive cell cycle analysis of the CENP-C complex extracted from cells by western blot and AFM analysis. We also expanded our analysis using CENP-A mutants.

Previously we used a CENP-C antibody (PD030) in combination with ACA serum for serial nChIP. The original CENP-C antibody is no longer available, forcing us to find a new CENP-C antibody that would pull down CENP-C that was applicable for both Western blot analysis and AFM analysis. Indeed, we found a new CENP-C antibody (EPR15939). Furthermore, we have since developed a custom CENP-A antibody. This allowed us to compare our first set of antibodies with the new set of CENP-C and CENP-A antibodies. We found very similar results between the two sets of antibodies, which further supports our original claims that two CENP-A populations exist in cycling HeLa cells.

In addition, we expanded our analysis to test the dynamics of the two CENP-A populations and the CENP-C complex across the cell cycle. By Western blot analysis and AFM analysis, we found that the two CENP-A populations are present across the cell cycle. The CENP-C complex had a stable Feret's diameter, but its height increased as the cells progress into mitosis.

Finally, we expanded our interrogation of the interaction between the two CENP-A populations and the two CENP-A mutants. We established that the two CENP-A mutants form nucleosomes *in vivo* and localize to the centromeres. By Western blot analysis we were not able to show that the H3^{CpA} CTD mutant a strong interaction with CENP-C (native or overexpressed). This complicated our initial interpretation and thus we have updated our manuscript to reflect this uncertainty. We have toned down our discussion as well to accurately reflect our findings.

We hope the reviewers can appreciate the new work to strengthen our original manuscript. Below we responded to each individual point brought up by each reviewer.

All the figures have been updated, including the addition of new Figures 5 and 6. For the reviewers' convenience, we have made all textual changes in the main text blue.

Sincerely,
Daniël Melters and Yamini Dalal

Reviewer #1 (Comments to the Authors (Required)):

The manuscript #LSA-2024-02819 aims at characterizing centromere-specific chromatin using nanoscale imaging techniques and biochemical approaches to better document the hierarchy of protein complexes, among which are the kinetochore and the constitutive centromere associated network (CCAN), assembled at centromeric loci.

At centromeres, it has been well-documented that CENP-A provides an epigenetic and structural foundation for the recruitment of inner kinetochore proteins, among which is CENP-B, which in turn recruits the CCAN. The project that forms the core of this manuscript stemmed from the observation by others of a large and compact macromolecular complex identified through immunoprecipitation of native CENP-A chromatin (nChIP).

The authors then decided to tackle the question of the identity and structure of this never-studied macro-complex.

Here, they revealed that the CENP-C complex is a large polygonal structure that is physically associated with four to six CENP-A nucleosome and contains CCAN members. Surprisingly, this complex associates with only a small fraction of the total nuclear CENP-A with a large fraction of CENP-A nucleosomes not being associated with CENP-C. Their data suggest that this large complex represents the inner kinetochore/CCAN with nucleosome-free DNA organized by CENP-C. They also attempted to link a CENP-C overexpression context to increased centromeric transcription and mitotic defects, and its rescue by CENP-A mutants, although this part and the rationale for it are the least convincing.

As a whole, the manuscript is interesting for the centromere community and is based on rather solid and convincing imaging at the nanoscale. The Development of the immuno-AFM approach is also new and very interesting.

However, I found the whole work difficult to follow. Compared to their current MS in BioRxiv, part of which has been published in eLife 2023, the present MS is far less well written and described, suffers from expeditive writing and conclusions and is missing a clear rationale for studies in several places. The redundancy with their previous manuscripts (Melters 2019 and 2023) is also somewhat disturbing. To finish, I am afraid that the claim that the data support the model of "a balance between unbound and kinetochore bound CENP-A chromatin is needed for supporting a unique higher-order structure at centromeric chromatin" remains rather weak at this stage. Hence, I would suggest major modifications throughout the text.

Response: We thank the reviewer for their accurate summary and overall positive take. We have added new experimental data, resulting in an extensively revised manuscript, including new and updated figures accompanied with all necessary textual changes. We added biochemical and biophysical analyses of the CENP-C complex and associated nucleosomes throughout the cell cycle. We also further tested how the CENP-A mutants modified the two CENP-A populations. These results are summarized in Figure 4-7. In addition, we have added graphics to guide the reader through the experimental approaches. The latter set of experiments led us to tone down our model to match our

findings. We are confident that we have made significant improvement to our manuscript.

The manuscript should be improved in terms of writing, especially for the description of the approaches used in the results section so that the reader does not need to search the information in the methods section and in the legends to figures.

Response: We acknowledge the reviewer comment. We have not only rewritten the results section, but we have also added graphics to the figures to help guide the reader through the experiments.

The abstract does not seem to really match conclusions drawn from the work:

- CENP-C complex-bound chromatin has strongly reduced DNA accessibility?
- the CENP-C complex matches the dimensions of both a single and multiple CCAN?

Response: In light of the addition of new experimental data, we have updated the abstract accordingly. As to the first point, MNase is an aggressive nucleic acid nuclease and in Figure 1E, we show that MNase has restricted nuclease activity compared to bulk or CENP-A nChIP chromatin.

There are many places that need further clarification - Examples follow:

Response: We have taken this comment to heart and as such we have revised many parts of the manuscript, using both comments from all three reviewers and also based on our own rereading of the manuscript.

line 102, "therefore": what is the link between "small fraction of macromolecular complexes within CENP-A native ChIP » and developing an assay to purify CENP-C-bound chromatin?

Response: This section has been heavily revised. The logic that led our interest in pursuing the isolation and imaging of the kinetochore, was driven by the observation that previously we occasionally observed large complexes in CENP-A nChIP. At that time, we were unsure what these complexes were. Whether or not these were artifacts. Once we developed CENP-C nChIP that produced analyzable product for AFM, Western blot analysis, electrophoresis analysis, were we able to identify and study these large complexes.

Paragraph "CENP-C complexes are diverse in size":

- an unclear title since the paragraph deals with the large polygonal complexes that came down with CENP-C nChIP. What characterizes the different complexes at this point?

Response: We have updated the title of this section to: "CENP-C complex dimension across the cell cycle".

- Please precise what definition you used for the Feret diameter. The whole calculation is a little bit convoluted. How a surface with 15-17 nm sides can make a circle of 22 nm? Why it "would have" a height of 9.3 nm?

Response: We consistently used the maximum Feret's diameter when referring to the Feret's diameter. We have clarified this in the manuscript (lines 248-252). The dimensions mentioned by the reviewer are derived from the cryo-EM structure shown in Figure 5A.

- Apparent contradiction between "CENP-C complexes were associated with... nucleosome-free DNA" (line 117) and "CENP-C can... potentially associate with up to four CENP-A nucleosomes" (line 142) and also "CENP-A nucleosomes are indeed associated with the purified CENP-C complexes" (line 182).

Response: We are of a different opinion as the reviewer with regard to the "contradiction" of nucleosomes and nucleosome free DNA. Nucleosome free DNA and nucleosomes are commonly found across the genome. Promoters of highly transcribed genes are commonly nucleosome free, yet nucleosomes are known to be phased over the 5' end of a gene. Here, we observed both nucleosomes associated with the CENP-C complex, and we observed stretches of naked DNA associated with the CENP-C complex. In Figure 2C, D, and 5B nucleosomes and naked DNA can be clearly seen in association with the CENP-C complex.

- What is a stepwise correlation in this context? (line 143 and line 146-147).

Response: We have extensively updated the manuscript. With the cell cycle analysis of the CENP-C complex, we observed no change in Feret's diameter

Line 217, "there is a ~6-fold excess of CENP-A that is not strongly associated with CENP-C": precise if nucleosomal CENP-A?

Response: This statement is no longer present in the manuscript. Instead, we have expanded our nucleosomal analysis of bulk chromatin, CENP-A nChIP, and CENP-C nChIP associated nucleosomes to across the cell cycle. See Figure 4B.

Methods: PacBio sequencing would deserve more information: how many reads, length of fragments, QC, alignment, which "sequences in Repbase"? etc...

Response: Thank you for this comment. We have updated the methods section accordingly.

What is the logic of confronting "a critical balance must be maintained between kinetochore-bound CENP-A chromatin and kinetochore-free CENP-A chromatin" with "a CENP-C overexpression scenario"?

Response: We have extensively revised this previous manuscript, and we have included new experimental data testing the interaction between the two CENP-A populations and the two CENP-A mutants. The results we obtained are not in agreement with our original hypothesis (Suppl Figure S6). The mitotic defect corrections induced by the two CENP-A mutants in the CENP-C overexpression scenario imply a link between centromeres and centrosomes. As elucidating this link goes beyond the scope of this manuscript, we limited ourselves to a section in the discussion.

Lines 285-287: predictions should be experimentally tested and experimentally validated

Response: See our response above regarding the two CENP-A mutants experiments.

Lines 289-290: Explain the rationale for looking at centromeric transcription

Response: Previously we showed that centromeric transcription is reduced in the CENP-C overexpression background and that this reduction is accompanied with impaired loading of new CENP-A (Melters et al 2019; 2023). That is why we wondered if the CENP-A mutants could restore this defect. Indeed, we show in Figure 7B that the CENP-A mutants restore centromeric transcription.

Line 291, "a ~60% reduction in a-satellite transcription": what are the basic levels in Hela cells? Please comment on the fact that centromeres are supposed to be repressed in "normal" cells (control set at 1, so is not informative). This sentence is also found in their previous paper, so please quote your work when it is redundant with previous papers.

Response: We like to note that centromeric transcription is a common feature in all species tested thus (reviewed in Arunkumar & Melters 2020). This is not to say that centromeres are transcribed at levels similar to house-keeping genes. We and others have noted that centromeric transcription happens at a low level compared to house-keeping genes. That said, we have added additional statements about how the transcript levels were normalized (we used GADPH) and we cited our previous.

Please provide controls for H3CpA CTD or CENP-A Δ CTD ectopic expression. What is their localization in transfected cells or mitotic chromosomes? Please provide controls for "CENP-A mutants that function either as a sink or are unable to bind CENP-C".

Response: To address this concern, we have added IF images showing that the CENP-A mutants colocalize to mCherry-CENP-A at the centromere (Figure 6D).

Line 301: to what relate (40% normal, 60% abnormal) and (74% normal, 36% abnormal).

Response: Thank you for pointing out this lack of clarity. As such we have added a sentence to clarify what abnormal mitosis phenotypes we scored.

Line 306: the "rescue" of mitotic defects is not convincing as there is no statistical analysis and lagging chromosomes remain virtually unchanged. This should be discussed.

Response: We have softened our statement of "rescue" to "partial rescue" to further highlight what we already wrote. Multipolar spindles and multipolar spindles + lagging chromosomes were commonly found in the CENP-C overexpression and these in particular were rescued by the CENP-A mutants. We also added a brief discussion about this finding in the discussion.

Lines 309-315, this concluding paragraph raises some questions:

- What happens to endogenous CENP-C upon CENP-C ectopic expression? Same for CENP-A. What is the localization of mutants CENP-A?
- In these conditions, the authors should assess the impact on kinetochore-bound and kinetochore-unbound CENP-A chromatin

Response: We thank the reviewer for these questions. Previously we showed that CENP-C overexpression does not impact CENP-C levels, but it does result in reduced CENP-A levels (Melters et al 2023). Indeed, here we didn't observe a noticeable change in endogenous CENP-C levels upon CENP-C overexpression. In the H3^{CpA} CTD mutant, endogenous CENP-A levels were not impacted, but in the CENP-A^{dCTD} mutant, endogenous CENP-A levels were reduced (Figure 6C). Interestingly, when we overexpressed CENP-C together with the CENP-A^{dCTD}, we observed that the CENP-A levels were unaffected (Figure 7A). We have updated the results and discussion section to reflect these new findings.

The discussion should be revised in the light of previous comments.

Response: As the reviewer might appreciate, the entire manuscript has been heavily revised, including the discussion.

Figure 3 is puzzling at first sight because the panel in column 2/raw 2 is not at the same scale as the other images.

Response: We apologize for the confusing scale bars. We have updated Figure 3A to make sure that all the AFM images and scale bars are set to the same size.

Reviewer #2 (Comments to the Authors (Required)):
General summary & opinion:

In the present study, Melters et al. analyze the structure of chromatin-bound CENP-C in human cells. For this, using the method described in their previous studies (Dimitriadis et al 2010, Walkiewicz et al 2014), they performed native chromatin-immunoprecipitation (nChIP) of CENP-C in HeLa cells. They further pursue their analysis using single molecule imaging (as described in their publication Melters & Dalal 2021) and biochemical approaches. They found that CENP-C associated with a large complex bound to CENP-A, a complex more resistant to nuclease digestion than bulk or CENP-A containing chromatin. This complex could correspond to a kinetochore-bound complex associated with CENP-A, with reduced DNA accessibility. Using serial nChIP with first CENP-C followed by ACA antibodies, the authors also revealed two possible CENP-A nucleosomal configurations: (i) a small portion associated with CENP-C and (ii) the vast majority remaining, not associated with CENP-C. Finally, authors performed CENP-C overexpression experiments. This overexpression led to 60% reduction in alpha-satellite transcription, similarly to previous result (Melters et al 2023), along with mitotic defects. These defects were rescued by the introduction of CENP-A mutants. Overall, the authors conclude that based on CENP-C association, two populations of CENP-A-nucleosomes co-exist in human cells. They further propose that a proper balance between these populations would be crucial for proper centromere function:

- The CENP-C unbound population would maintain centromeric transcription and CENP-A levels at centromeres,
- The CENP-C bound population (along with kinetochore proteins) would ensure faithful chromosome segregation.

CENP-A nucleosomes display specific features including looser DNA binding and increased dynamics of DNA ends (Sekulic et al 2010, Panchenko et al 2011, Tachiwana et al 2011, Lacoste et al, 2014, Ali-Ahmad et al 2019; Arimura et al 2019). But how this local flexibility and CENP-A interactors together impact higher-order structure of centromeric chromatin remained unclear. These specific features might lead to a dynamic structure and may also involve compact structures (Ali-Ahmad et al 2019, Melters et al 2019, 2023). Importantly, within the CCAN how CENP-A nucleosomes along with their distinct structures are placed is still enigmatic. In this context, this manuscript, which represents a follow-up of previous studies from the same authors (Melters et al 2019, 2023) is relevant. It confirms the existence of two types of CENP-A nucleosomal complexes as hypothesized in previous studies (Henikoff et al 2015, Kyriacou & Heun 2018). Importantly, to document the importance/role of these two types of structures would represent a valuable contribution for the field. However, the second part of the study deserves to be revisited to firm up the conclusions. Importantly, series of concerns remains as listed below in detail, along with suggestions to help to address them and better substantiate the conclusions. These points should be considered carefully before publication.

Response: We thank the reviewer for these kind and thought provoking comments. We have taken these comments to heart, and we have tried to address all the concerns.

Major concerns:

(1) A main concern relates to their final model which argues that the balance between the two CENP-A fractions is crucial for centromere function. To draw this conclusion, additional experiments should be performed:

- First, to confirm that the overexpressed CENP-C actually binds CENP-A chromatin and change the balance between CENP-A free and bound-CENP-A nucleosomes (saturates CENP-A nucleosomes).
- Second, to provide data on whether the rescue with CENP-A mutants does restore a proper balance between CENP-C-free and -bound nucleosomes.

[Response: We thank the reviewer for suggesting these experiments. We have added biochemical, AFM, and IF experiments to address the concerns raised here. First, we expressed the two CENP-A mutants in HeLa cells without overexpressing CENP-C and found that HA-H3[^]CpA CTD did not come down with the CENP-C nChIP and only a small amount of CENP-C came down with HA nChIP. In contrast, HA-CpA[^]dCTD did not come down with CENP-C nChIP as expected. Surprisingly, native CENP-A levels were dramatically reduced in the latter mutant \(Figure 6\). The mutants did form nucleosomes and colocalized to centromere using mCherry-CENP-A as a marker. When we overexpressed CENP-C in combination with the CENP-A mutants, we did see that CENP-C came down with second CENP-A nChIP. The second nChIP was also able to pull down excess CENP-C and native CENP-A levels appear to be restored in the HA-CpA[^]dCTD mutant. These results draw a more complicated picture how the mutants restore centromeric transcription and correct multipolar spindle mitotic phenotype in the CENP-C overexpression background. Multipolar spindle errors are commonly caused by centrosome defects. These results imply a potential crosstalk between centromeres \(CENP-A chromatin: CENP-C\) and centrosomes. . As such, we](#)

have toned down our initially proposed model. As we expanded our study also with an extensive cell cycle characterization of the CENP-C complex and CENP-A nucleosomes (see below for more detail), we believe that understanding how centromeres and centrosomes crosstalk is beyond the scope of this manuscript. We discuss this latter idea in the discussion.

(2) The serial nChIP (CENP-C & ACA) in asynchronous cells revealed two CENP-A nucleosomal configurations is an interesting observation. The authors could comment on the choice of ACA serum, instead of CENP-A antibody. To strengthen this finding, a CENP-A nChIP should reveal these two fractions, bound or not by CENP-C.

- If the authors had already performed this experiment in a previous study, they should make sure that the information is made available (referring to precise figures/experiments in the paper). Otherwise, this should be documented in this manuscript.

Response: we have added a sentence how our current results compare to our previous results.

- Possibly, an approach with synchronized cells would enable to discard possible differences reflecting possible cell cycle variations such as cell cycle coupled centromeric chromatin remodelers, or ectopic CENP-A assembled onto chromosome arms in early G1 (Nechemia-Arbely et al 2019).

Response: We thank the reviewer for these insightful comments. We have tried to address these concerns in two-fold. First, we used a second CENP-C antibody in combination with custom made CENP-A antibody. Using this new combination of CENP-C and CENP-A antibody, we observed the presence of two CENP-A populations, just as we did with the first CENP-C antibody in combination with ACA. Second, we compared the CENP-C complex (Figure 5), the two CENP-A populations (Figure 4A), and the nucleosomal dimensions (Figure 4B) across the cell cycle. Overall, we believe we now provide more robust data that two CENP-A populations exist in cells and that the CENP-C complex changes a little throughout the cell cycle (height increases, whereas Feret's diameter remains the same).

(3) The authors predict that CENP-A mutants function as sink for excess CENP-C (H3CpA-CTD) or increase the kinetochore-free CENP-A population (CENP-A Δ CTD), page 13. To support the following conclusions, the study should confirm that H3CpA-CTD mutants bind CENP-C, and that CENP-A Δ CTD mutants bind centromeric chromatin. Eventually, an IF analysis of CENP-A and centromeric markers in CSK extracted cells should provide more information on the effect of the CENP-A mutants.

Response: We thank the reviewer for these insightful comments. We have expanded our work with various lines of experimental data (Western blots, IF, and AFM) addressing the localization and impact on CENP-A populations in the presence of the two CENP-A mutants. In the H3^{CpA} CTD mutant pull down, we observed only very small amount of CENP-C come down, whereas in the first CENP-C nChIP, we did not observe any of the mutants being pulled down, either under endogenous CENP-C levels (Figure 6C) or CENP-C overexpression conditions (Figure 7A). Interestingly, in the CENP-A^dCTD mutant, endogenous CENP-A levels were reduced (Figure 6C), but in the CENP-C overexpression condition, we observed that the CENP-A levels were

unaffected (Figure 7A). We have updated the results and discussion section to reflect these new findings.

Series of controls or specifications are also missing:

(1) In figure 5D, important markers should be added to control the ChIP-Western analysis:

- The HJURP staining should be added.
- The CENP-B staining, mentioned in the text (line 236), is missing in the figure and should be added too as it is an important control (as a CCAN member and one of the targets of the ACA antibody used).

Response: We have added both CENP-B and HJURP Western blot staining to what was Figure 5D and is now Figure 1D. As the reviewer can see, CENP-B is found in both CENP-C and ACA nChIP, whereas HJURP is only found in CENP-C nChIP. We have updated the text and methods section accordingly.

(2) In figure 1C, the legend should indicate for the 3D rendering to which conditions it does correspond to. In addition, the figure should show 3D rendering images for both CENP-A and CENP-C pull downs (with the height scale) for comparison, and not only one condition.

Response: We have updated Figure 1 (not Figure 2) to also include a graphic explaining how the experiment was done, arrows to help guide the reader, and 3D rendering of bulk chromatin, CENP-A nChIP, and CENP-C nChIP, including labels.

(3) The manuscript should indicate the levels of CENP-C overexpression, upon transfection, compared to normal expression levels. This is important to discuss the issue of balance. When does the tipping over occur? Is there a threshold?

Response: We thank the reviewer for pointing this out. CENP-C is roughly 2-fold overexpressed. We have added this statement to the manuscript. Furthermore, as we have toned down the statement due to the findings of the CENP-A mutant experiments (Figures 6 and 7). Although we do believe a balance exists, with the updated data in this manuscript, this would be too speculative to argue this point.

(4) Same comment about the CENP-A levels upon addition of CENP-A mutant constructs. To be able to compare the results, it is important to ensure that comparable levels of expression have been achieved.

Response: The H3^{CpA} CTD mutant was incorporated into chromatin at ~27% of endogenous CENP-A based on HA nChIP. The precise levels of CENP-A^{dCTD} compared to endogenous CENP-A was hard to measure, as endogenous CENP-A levels were reduced in this mutant (Figure 6C). Interestingly, in the CENP-C overexpression background, the presence of CENP-A mutants did not impact the levels of endogenous CENP-A (Figure 7A).

- Legend of suppl. Table S4 mentions the quantification "from Western blot" of CENP-A levels in the nChIP fractions but the result is not shown. This data should be added to

the manuscript, along with the total CENP-A levels in whole cell extracts in each condition.

Response: we apologize for the confusion. The quantifications shown in Suppl Table S4, now Suppl Table S1, refers to the quantification shown in Figure 1C. We have updated the text to clearly reflect this.

• In parallel, for establishing their model, the authors should consider the impact of high CENP-A levels on its localization (Lacoste et al 2014), and the fact that ectopic CENP-A can recruit CENP-C at ectopic regions (Van Hooser et al., 2001; Gascoigne et al., 2011; Lacoste et al., 2014).

Response: We thank the author for bringing up this concern. Previously we showed that (Athwal et al 2015) that CENP-A levels in HeLa cells were relatively similar to primary cells (Figure 1A in Athwal et al 2015) and few ectopic sites were found (Figure 7 in Athwal et al 2015). In addition, the CENP-A mutants we used in this study did not result in a massive overexpression of endogenous CENP-A or CENP-A mutants (Figure 6C and 7A). We therefore find it very unlikely that ectopic CENP-A levels would be a significant factor in explaining our observations.

(5) The CENP-A mutant constructs should be added to the manuscript. The experimental protocol to introduce the "CENP-A mutants" in cells is actually missing in the material & methods section of the manuscript and should be provided.

Response: We have updated the methods section to include how cells were transfected. We have also submitted our construct to Addgene to make them readily available to the field.

(6) Figure 8A should indicate what is revealed (the antibodies/colours used). The scale bar values should be added, at least in the legend. The figure should also mention the condition in which condition the images were taken. In addition, there should be representative images for each of the 4 conditions (WT/Mock, CENP-C OE, the 2 rescue experiments).

Response: We have updated what was Figure 8A and is now Figure 7C to show representative images of the four conditions, including individual panels for CENP-C and tubulin staining and a merge with DAPI. Arrows have added to the figure to indicate lagging chromosomes.

(7) CHIP & FISH data have shown that centromeric CENP-C levels are lower compared to those of CENP-A (Henikoff et al 2015, Kyriacou & Heun 2018). The mapping of CENP-C and CENP-A on APEX-chromatin fibers in Kyriacou & Heun 2018 showed that CENP-C is organized differently from CENP-A (a fraction of CENP-C localizes at places where CENP-A is absent) inside the centromeric domain. This is in line with the first conclusion of the study, and the authors should mention/discuss these previous reports.

Response: We thank the reviewer for pointing these studies out. We are acutely aware of these findings and in a previous version these findings were indeed discussed as the reviewer pointed out. We have added this discussion back.

Minor corrections & additional comments or questions:

(1) Could you detect micronuclei, following cell division? And do the cells keep cycling/proliferating as in the control condition?

Response: Thank you for these questions. We perform IF with the mutants while the cells were in mitosis. To observe micronuclei, we would need to check the cells post-mitosis. We have an ongoing project in our lab looking at micronuclei. Cells that overexpressed CENP-C not only had many mitotic defects. These cells were generally unhappy, including lower proliferation rates.

(2) Figures 5/4/3/7: Please indicate in the legend what the error bars correspond to (even when it appears in the table of the supplementary data).

Response: Although the figure numbering has altered, we have updated all the legends to clarify that the error bars correspond to the standard deviation.

(3) In figure 2D, what does each point of the scatter plot represent? The legend should indicate if each dot represents a single or averaged measure...

Response: The legend has been updated as requested.

(4) Figure 4 should be refined. In addition, it could possibly be part of figure 3...

- The plots seem to represent the height of CENP-A nucleosomes measured by AFM in the two fractions (CENP-C nChIP followed by CENP-A nChIP). The legend should be reformulated to indicate all the information clearly (height of CENP-A nucleosomes measured by AFM, in the CENP-C nChIP or in the CENP-A nChIP that follows, with the same anti-CENP-A & secondary antibodies as in figure 3).

- The legend mentions 4 conditions whose measures would be quantified in the figure: Bulk chromatin, CENP-A nucleosomes pulled down by CENP-A NChIP, CENP-A nucleosomes pulled down by CENP-C nChIP and "in addition ... large structures that came down by CENP-C NChIP". There are only 3 box plots and thus, it is unclear what condition is measured for each box plot.

- The figure should show representative images for each condition measured.

Response: As the reviewer can appreciate, all figures have been heavily revised, including the height measurements of nucleosomes. The nucleosomal height measurements of asynchronous cells are now part of Figure 3C. The nucleosomal height measurements across the cell cycle can be found in Figure 4C. Representative images have been added to all AFM measurement figures.

(5) Figure 5: The authors could add the molecular weights on the Western blot analysis.

Response: We thank the reviewer for this suggestion. As the figures are already complex, we try streamline the data presentation for clarity as much as possible.

(6) Figure 7: The name of the legend is incorrect: It is the same as in figure 8.

Response: We thank the reviewer for pointing this error out. All the figures have been heavily revised and the corresponding legends have been adjusted accordingly.

(7) In figure 8B, it is not mentioned what the "n" counts refer to precisely. In addition,

quantification of mitotic defects should be represented more precisely, with bar plots indicating error bars for the quantification.

Response: We thank the reviewer for this comment. We have updated the mitotic defect figure, which is now Figure 7C, D.

(8) Line 25: Sources should be indicated for the following sentence: "Inaccuracies in kinetochore assembly can lead to the formation of dicentric chromosomes, or chromosomes lacking kinetochores. In either case, chromosomes fail to segregate faithfully, which drives genomic instability."

Response: Thank you for pointing this out. Interestingly, before reading this comment, we had already added references. Our apologies for the omission.

(9) Line 31-32: their statement: "At the base of the inner kinetochore is the histone H3 variant CENP-A (CENH3), which provides the epigenetic foundation to assemble the kinetochore, recruiting inner kinetochore proteins CENP-B, CENP-C and CENP-N." relate here to a human system, thus it would be important to specify the following points.

- The sources Régnier et al and Mendiburo et al are indicated, the model used (chicken, Drosophila) for each publication should be specified.
- In addition, since this study relies on human cells as a model, sources for human cells could be added (Carroll et al 2009, 2010, Allu et al 2019).

Response: We thank the reviewer for pointing out these additional references, which we have subsequently cited, including the addition of a statement referencing model species.

(10) Line 109: "we observed large polygonal structures, with a roughly circular footprint". The authors could indicate the % of appearance of this type of structure. In other terms, did this pattern (shown in figure 1 and in representative images of figure S1) appear in 100% cases (177 out of the 177 measures) or at another, very high frequency?

Response: We thank the reviewer for this comment. We have added the quantification of the roundness factor of the CENP-C complex in Figure 5F, as well as Table 1. As the reviewer can appreciate, from S phase through M phase, the roundness remains the same.

(11) Line 150: "The precise composition of individual native CENP-C complexes is unknown, but mass spec analysis of tagged CENP-A nucleosomes brought down both kinetochore and non-kinetochore proteins (Foltz et al, 2009; Roulland et al, 2016; Remnant et al, 2019)." The reference Dunleavy et al 2009 should be added since it showed by immunostaining how, upon HJURP downregulation (and subsequent loss of centromeric CENP-A), CENP-C association with centromeres is decreased by contrast with CENP-B, for example.

Response: We have added the reference Dunleavy et al 2009 as requested.

(12) Line 241: The "Purified CENP-C complex robustly represents the CCAN components". To a lesser extent, this seems also true for the ACA-targeted (CENP-C-free) complex. The corresponding figure (5D) should be completed with the

quantification of the Western blot, and the sentence should be phrased with a little caution (e.g., something like "CCAN components are more enriched in the purified CENP-C complex than the CENP-C-free complex, indicating that...")

Response: We thank the reviewer for pointing this out. This sentence has been heavily edited during the revision of this manuscript. The overall tone of the manuscript has been toned down to be more cautious.

(13) Line 359: "In many solid tumors, CENP-A is overexpressed and found outside the centromere by high-jacking H3.3 chaperones (Lacoste et al, 2014; Athwal et al, 2015; Zhao et al, 2016; Shrestha et al, 2017, 2021; Nye et al, 2018; Arunkumar et al, 2022) or by altering the H3-H4 histone supply (Balachandra et al, 2024)." Although CENP-A is frequently overexpressed in cancer, its mislocalization has actually only been documented in cell lines. To be accurate, the sentence should be rephrased to highlight this important link between CENP-A overexpression in tumors and cell lines and CENP-A mislocalization in cell lines, along with the work of Verrelle et al. (2021), which gives hints in terms of changes in CENP-A subnuclear distribution in normal and tumoral tissues.

Response: The discussion has been heavily revised, and this paragraph has been removed for a more streamlined discussion section.

(14) Line 366: "CENP-A is preferentially found at genomic sites of high histone turnover..." The reference Lacoste et al 2014 could be added.

Response: We concur that this reference is should have been added, but during the revision of the manuscript, the discussion has been streamlined and this paragraph was removed.

(15) Line 368: We suggest rephrasing the sentence for clarity purposes, for example: "The two following questions ensue: first, what is the mechanism by which lncRNA recruit new CENP-A nucleosomes, and second, what is the role of CENP-C in this process?"

Response: The discussion has been heavily revised, and this paragraph has been removed for a more streamlined discussion section.

(16) The section "Materiel and Methods" mentions the CENP-A antibody used for NChIP figure 1 was "custom made". If possible, the authors should provide a reference or an immunostaining and data further confirming the specificity of the antibody.

Response: We thank the reviewer for this comment. In the main text, we reference papers that validate the antibody.

(17) The section "Materiel and Methods" describes protocols for a double thymidine block & for a quench-pulse-chase experiment. These experiments do not appear in the main manuscript. The experimental procedures that were not used in the study should be removed.

Response: We apologize for these errors. The double thymidine block experiments have been added and the quench-pulse-chase protocol has been removed.

(18) Figure 2B should indicate from what experiment the image comes from (xChIP, nChIP...)

Response: All figures have been updated and the type of ChIP is reported where applicable, as well as which antibody was used.

Reviewer #3 (Comments to the Authors (Required)):

This manuscript by Melters et al entitled seeks to answer an important and long-standing question regarding how centromeric chromatin conforms when bound to CENP-C, one of its direct binding partners. Here, they purify centromeric chromatin and view these purified constructs by AFM and TEM. They demonstrate some beautiful AFM of CENP-A nucleosomes bound to CENP-C. They identify both CENP-C bound and unbound CENP-A conformations, with CENP-C bound chromatin arranging into clusters of nucleosomes around a large polygonal (CCAN) structure, generally refractory to MNase digestion. The authors previously showed that CENP-C overexpression reduces transcription (Melters et al, 2023), and they correlate this with general inaccessibility of CENP-C-bound chromatin. However overall, I found this manuscript rushed, lacks attention to illustration and reader viewability, and would need significant improvement for my approval at this stage.

Response: We thank the reviewer for the kind words and acknowledge the critical notes. We have tried to improve the overall manuscript by adding experimental data, updating figures and improving the text.

Major Points:

1. Figure 1 provides one of the major advances in this manuscript - an ability to image CENP-C bound centromeric chromatin. This is an excellent achievement and I commend the authors here. This figure would substantially be improved by a schematic illustration of the experiment from the outset. It takes a lot of cross referencing with legends and text to figure out what each figure represents. Point features out and don't assume the reader knows what they are looking at. This figure needs to be viewed in tandem with Fig S1 so I suggest the same there. Please keep this in mind where relevant for other figures also.

Response: We thank the reviewer for the kind words and the suggestions to improve this Figure. We have added a graphic to show how the experiment was done, as well as colored arrows to point to specific features. We also applied the same logic to the other figures to help guide the reader to more easily understand the experimental set-up.

2. Figure 4 contains a measurement of nucleosome height. Can the authors include images of and point out these differing nucleosome heights? Some clear images of exactly what is being measured here, with plenty of examples, are absolutely required.

Response: Our apologies for the lack of clarity. We have done nucleosomal measurements for over 15 years using AFM, which is an ideal tool to make such

measurements as it is a topographical microscope where a tip scans the surface. We have added a sentence in the text (line 149) to clarify this. Furthermore, we have combined the immune-AFM and in vivo nucleosomal height measurements into a single figure (Figure 3). We have also added images using white arrows to point at clearly identifiable nucleosomes, which we measured.

3. There is a strikingly large subfraction of CENP-A not associated with CENP-C. However, I worry to what extent this is over estimated, or simply just non-centromeric CENP-A. FRAP studies have shown CENP-C to be quite dynamic throughout the cell cycle, with stability during S-phase and mitosis (Hemmerich et al 2008, JCB). In addition, the Fukagawa lab have shown that dynamic changes in CCAN organization during progression of the cell cycle (in DT40 cells). I worry many of these effects we see associated with CENP-C are a consequence of the cell cycle stage and overall dynamics - and only the stable CENP-A/CENP-C interactions are being picked up. The diversity in nucleosome size in Figure 2 probably supports this.

All of these experiments have been done using asynchronous HeLa cells. I feel this manuscript would be significantly more informative with a cell cycle characterisation of this centromeric chromatin. This is likely to change throughout the cell cycle up until "final" point through mitosis. How these CENP-C-bound CENP-A chromatin look under AFM/TEM at different points of the cell cycle is worth looking at and might clear up some variability. In addition, we already know that CENP-A gets seeded ectopically on chromosome arms at some frequency and is not kinetochore bound. Other very recent publications from the lab make reference to this (e.g. Bui et al, 2024). Nechemia-Arbely et al, (Nat Cell Bio, 2019) suggest an error correction mechanism during S-phase whereby CENP-C protects centromeric CENP-A through replication, and ectopic CENP-A gets removed. If the vast majority of your cells are in G1, this may also be the reason you have a large majority of unbound CENP-A. This should be cleared up.

Response: We thank the reviewer for pushing us to look at the CENP-C complex and associated nucleosomes throughout the cell cycle. All this work has been summarized in Figures 4 and 5 and updated Table 1. The original anti- CENP-C antibody we used has been discontinued and in the meantime we obtained a new custom made anti-CENP-A antibody. As a result, we now show that, using two different sets of CENP-C and CENP-A antibodies, throughout the cell cycle we pull down two CENP-A populations. The nucleosomes that came down with CENP-C nChIP remain roughly the same height, whereas we replicated our finding that nucleosomes that came down in the CENP-A nChIP were taller in S phase compared to the rest of the cell cycle (Bui et al 2012). Furthermore, we don't see a cell cycle dependent change in Feret's diameter for the CENP-C complex, but the height of the CENP-C complex increased throughout the cell cycle and was the tallest in mitosis.

5. Figure 5 contains a sequential NChIP assay. However, ACA does not strike me as the appropriate antibody for the second NChIP in this case. Using a serum that contains CENP-A, CENP-B and CENP-C together is possibly not the cleanest approach. Can the authors explain their reasoning here? The fact that CENP-C complex associates with the CCAN kinetochore supports what this circular structure in the AFM images might be, and I agree it probably is based on Fig 5D. In line 237/238, the authors say that HJURP

was enriched in the CENP-C NChIP. No data in main or supplemental figures show this, nor is it referenced in the text that I can see. Please include this data.

Response: We agree with the reviewer that ACA is not the ideal antibody to only pull down CENP-A. At the time these experiments were started, we did not have an antibody to efficiently and reliably pull down CENP-A. We have developed such an antibody since, and we have used this in our most recent publication (Bui et al 2024). As the CENP-C antibody we initially used is no longer being made, we had to switch CENP-C antibody that is also ChIP-Western and ChIP-AFM grade. As a result, all the cell cycle and CENP-A mutant experiments were done using the new CENP-C antibody and our custom CENP-A antibody.

6. The idea of expressing CENP-A mutants that can either sequester away excess CENP-C or which are insensitive to CENP-C overexpression is an interesting and entirely plausible hypothesis. The authors previously showed that CENP-C overexpression significantly reduced alpha-satellite transcription (Melters et al, Elife, 2023). The qPCR data in Fig 7B restates this and includes their mutant lines that restore expression levels. This is generally well-executed. How can the authors reconcile the misregulation of centromere transcription in many cancers, with the known fact that CENP-C is not overexpressed in solid tumours (per their discussion)? Overall I see Figure 7 as an opportunity missed without further exploration. After going to the trouble of making these mutants, why not put these constructs into their AFM/TEM assays? This may give you a more information on how CENP-C influences chromatin organisation. The H3CpA CTD mutant would be particularly interesting, and worth including.

Response: We thank the reviewer for this interesting direction of inquiry. We expanded our work by including Western blot, AFM nucleosome analysis, and IF experiments (see Figures 6 and 7). Our hypothesis that the two CENP-A mutants we made (H3^{CpA} CTD and CENP-A^{dCTD}) would rescue CENP-C overexpression-induced defects by either serving as a CENP-C sink (H3^{CpA} CTD) or create a centromeric CENP-A domain that cannot bind CENP-C (CENP-A^{dCTD}). We first expressed the mutants in WT HeLa cells and were unable to show that CENP-C and H3^{CpA} CTD efficiently co-IPed. When we co-expressed the CENP-A mutants and CENP-C, we saw an increase in CENP-C coming down with the second CENP-A nChIP, but we did not see either of the mutants come down with the first CENP-C nChIP. These results contradict our initial hypothesis. As such we have increased our focus on the cell cycle dynamics of the CENP-C complex.

8. In Figure 8, the primary mitotic defect (lagging chromosomes), has not been rescued. The multipolar spindle defects are the predominant rescue (as the authors mention). Can the authors explain this and why they think this is important? Why would multipolar spindles be the only phenotype that this would rescue? As things stand, I don't find this figure particularly informative.

Response: We thank the reviewer for these questions. We are indeed intrigued by the observation that the CENP-A mutants do not rescue the lagging chromosomes but do rescue the multipolar spindle errors. As multipolar spindles are primarily caused by centrosome defects, we speculate that our results hint at a possible interplay between

the centromere and centrosomes, driven by CENP-C bound to CENP-A chromatin. What this interplay is, remains elusive and unfortunately is beyond the scope of this manuscript. We point out the possible interplay between centromeres and centrosomes in the discussion.

Minor Points:

1. Is it always only one large structure associated with CENP-A chromatin?

Response: As far as we have been able to see by TEM and AFM, we indeed only see one large structure per chromatin array. But as we extracted MNase digested chromatin, we do not believe we can say with any level of confidence that this reflects what is happening at centromeres. Single cell nChIP would be needed to get at this question, which is beyond the scope of this manuscript.

2. Does bulk chromatin differ substantially from CENP-A NChIP chromatin? It looks so based on TEM images in Fig 1 & S1.

Response: We are unsure what the reviewer means by differ substantially. The array sizes of bulk chromatin and CENP-A nChIP after MNase digestion are comparable (see Figure 1E, Suppl Figure S1). We updated the AFM image of CENP-A nChIP for Figure 2C to show that the arrays look similar.

3. Figure legend title error: Figure 7 and Figure 8 have the same title.

Response: As all figures have been updated, including the addition of new experimental results, we have updated all figure legends. This error has been corrected.

4. Fig 7, 8 and possibly 9 are quite descriptive and can be combined into one figure depending on revisions.

Response: Thank you for this suggestion. As the reviewer can appreciate, we have rewritten and updated all figures in the manuscript, including combining figures where feasible.

5. In the discussion (which is in general overly long), the authors discuss the potential of lncRNAs at the centromere. CCTT is another lncRNA that has been shown to directly bind and recruit CENP-C to the centromere (Zhang et al, Mol Cell). It makes sense that if CENP-C is so flexible, and binds multiple CENP-A nucleosomes, that a lncRNA might help stabilise this CENP-C interaction. This is worth citing in their discussion.

Response: We agree with the reviewer's assessment that the discussion was too long. Therefore, we have shortened the discussion substantially. We thank the reviewer for pointing out another very interesting paper. Unfortunately, we have cut the whole lncRNA section out of the discussion for a more streamlined discussion.

December 23, 2024

Re: Life Science Alliance manuscript #LSA-2024-02819R

Dr. Yamini P Dalal
National Institutes of Health
National Cancer Inst. (NCI)
41 Library Drive MSC 5055
Bethesda, MD 20892-5055

Dear Dr. Dalal,

Thank you for submitting your revised manuscript entitled "High-Resolution Analysis of Human Centromeric Chromatin" to Life Science Alliance. The manuscript has been seen by the original reviewers whose comments are appended below. While the reviewers continue to be overall positive about the work in terms of its suitability for Life Science Alliance, some important issues remain.

Our general policy is that papers are considered through only one revision cycle; however, given that the suggested changes are relatively minor, we are open to one additional short round of revision. Please note that I will expect to make a final decision without additional reviewer input upon re-submission.

Please submit the final revision within one month, along with a letter that includes a point by point response to the remaining reviewer comments.

To upload the revised version of your manuscript, please log in to your account: <https://lsa.msubmit.net/cgi-bin/main.plex>
You will be guided to complete the submission of your revised manuscript and to fill in all necessary information.

B. MANUSCRIPT ORGANIZATION AND FORMATTING:

Sincerely,

Reviewer #1 (Comments to the Authors (Required)):

The authors have put a lot of effort into revising their manuscript, both in terms of writing and experimentation.

It is now in a much better shape and will be of significant interest in the field.

The authors have responded convincingly to my comments, so I can only recommend publication.

Reviewer #2 (Comments to the Authors (Required)):

We thank the authors for their detailed answers and additional work along with the updates made to both the figures and the text.

First, the study documents the existence of two distinct CENP-A nucleosomal fractions in centromere chromatin: CENP-C-bound or not. This work reinforces previous studies (Henikoff et al 2015, Kyriacou & Heun 2018), including earlier work by the authors (Melters et al 2019, 2023). One concern that they addressed relates to possible changes during cell cycle. They have now included a cell cycle analysis of the two CENP-A fractions supporting their existence throughout the cell cycle, while also documenting the structures/complexes associated with each fraction. However, additional controls are missing to confirm cell synchronization in figure 4, such as IF images using EdU to label DNA synthesis and DAPI intensity to evidence enrichment in different cell cycle populations.

Second, to investigate the importance of the two distinct CENP-A fractions, the authors overexpressed CENP-C and used CENP-A mutant to "alter the binding landscape of CENP-C" (L318). The authors predict that CENP-A mutants could function as sink for excess CENP-C or increase the kinetochore-free CENP-A population (figure S5). Documenting the functional relevance of these two CENP-A fractions represents a valuable contribution to the field, but additional experiments were missing in order to confirm that CENP-A mutants restore a proper balance between the two CENP-A fractions. The authors now included biochemical and microscopy approaches to examine the CENP-A mutant constructs. However, to make a convincing argument, several issues remain unresolved and further improvement is needed for this critical part of the study as outlined below:

- (1) Biochemical analyses in figure 6 indicate that the HA-H3CENP-A-CTD does not pull-down CENP-C. In addition, incorporation into chromatin of CENP-A mutants remains questionable since neither H2A nor endogenous CENP-A co-IP with them. Therefore, there is doubt concerning functionality of these mutants and it is difficult to draw a strong conclusion regarding the importance of the balance between the two CENP-A fractions in chromatin.
- (2) Although the analysis suggests a trend, the CENP-A mutants do not restore significant centromere transcription, as shown in figure 7B, which contradicts the conclusion "CENP-A mutants restore centromeric transcription" (L356, L392...)
- (3) Figure 7C-D requires further attention. Percentages should be indicated in the bar plot, and a statistical analysis is missing in the analysis.
- (4) Finally, the authors adapted their model, now proposing that "CENP-C is critical in regulating centromere homeostasis by supporting a unique higher-order structure of centromeric chromatin and altering the accessibility of the centromeric chromatin fiber for transcriptional machinery" (L26...). This conclusion has already been proposed in earlier work by the authors (Melters et al 2023), thus the novelty of the contribution is now lost.

Other concerns:

- Fig. 3: Immuno-AFM targeting CENP-A indicates the presence of CENP-A in CENP-C complexes. In the text, the authors conclude about the "identity of the nucleosomes associated with CENP-C" (L179, L181, L1054...). However, the results do not support this conclusion and rather indicates the presence of CENP-A nucleosomes.
- For similar reasons, we recommend avoiding the use of the term "CENP-A ChIP" to describe the ACA nChIP which targets CENP-A, CENP-B and CENP-C (both in the text and figure legends).
- Fig. 6A: the mCherry tag should be displayed on the scheme.
- Fig. 6D: A CSK extraction prior to imaging could provide a good indication to determine if CENP-A mutants can get incorporated into chromatin (with respect to point 1).
- Fig. 6D: Centromere localization should be controlled through CENP-B or CREST staining.
- Fig. 6D: The DAPI is in blue and the GFP construct in green. Thus, in the top panel, the merged image shows a light, green-colored nucleus (mix of DAPI + mislocalized HA-H3CENP-A-CTD everywhere in the nucleus). However, in the bottom panel, the nucleus is also green (although GFP-CENP-AdCTD appears as green dots on the left image). Are the parameters and range of displayed pixel values the same between conditions?
- The title of figure 6 (L1066) is incorrect since the HA-H3CENP-A-CTD mutant does not localize to the centromere (as expected).
- L36-37: The two-layered structure of the kinetochore observed by electron microscopy was first showed in Robbins & Gonatas 1964 and described in Brinkley & Stubblefield 1966.
- L42: CENP-A is not the "foundation" for CENP-B binding, whose targeting is driven by a specific DNA sequence (Masumoto et al 1989) independently from the nucleosomes present (Carroll et al 2009).
- In L352, although CENP-A mutants form nucleosome sized particles and colocalize with mCherry-CENP-A, the conclusion "CENP-A mutants appear to form nucleosomes" is not sufficiently substantiated (with respect to point 1).
- Fig. 3 legend: how many assays and samples?

Minor formatting issues:

- L71: "a CENP-C complex" or "CENP-A complexes"
- L80: "from one another"
- L80: "the potential role of these two"
- L91: "In previous work (...), we noticed"
- L209: Regarding point 2, we would suggest changing the title (for example for "CENP-A nucleosomes associated with CENP-C are physically distinct")
- L221 ("we examined nucleosomes that are physically associated with the CENP-C complexes") is redundant with L210 ("we next turned our attention to individual nucleosomes associated with the CENP-C complex"), making it hard to follow/understand the objectives.
- L318: "we reasoned that introducing CENP-A mutants would alter"
- L368: "First, we performed"
- L402: "the most noticeable"
- L430: "Only one"
- L520: "were driven ... and were made"
- L612: "after treating the samples with proteinase K"
- L1024: "to the CENP-C complex"
- L1042: "done on the unbound fraction"
- L1072: "and the diameter of the mutants was determined using"
- Fig. 2B: The p-value is hidden behind a plot
- Fig. 3 legend: (A), (B) and (C) are misplaced, and (D) is missing
- Fig. 6 legend: the first sentence is unclear

Reviewer #3 (Comments to the Authors (Required)):

This manuscript has been substantially revised and focusses on the key technical advances. There is a significant improvement in how the manuscript flows and the layout of the figures. The authors took our advice to look at cell cycle stage and I believe this paid off. Whilst I am generally satisfied and I appreciate the effort the authors have put in to improve this manuscript since first submission, there lacks a key control for cell cycle stage. I cannot accept this manuscript without properly controlling for this. The timings suggested in Fig 4A look correct, but cell cycle stage shouldn't be assumed. The timings need to be shown to enrich these cell cycle stages. This can be shown by PI staining (DNA content) and flow cytometry, for example. i.e. what fraction of cells are enriched for each cell cycle stage at each time point? I am happy to accept pending this important addition.

Minor points:

- 1) Fig 1B: I recommend updating the blot for H2B here. Whilst it probably doesn't change the conclusion of the figure, it is worth getting a cleaner blot for H2B here given it is a main figure. It looks overexposed and the 2nd NChIP lane has a poor H2B band.
- 2) I would possibly suggest revising the CpC and CpA to simply CENP-A and CENP-A.

We thank the reviewers for their appreciation of our efforts to improve our manuscript both in experimentation and writing. To further improve our manuscript, we have added flow cytometry data to confirm each cell cycle stage (Supplemental Figure S5) and used an endogenous centromere marker to confirm centromere localization of the two CENP-A mutants (new Figure 6D). In addition, we have addressed all comments from each reviewer point-by-point below.

Sincerely,
Daniël Melters & Yamini Dalal

Reviewer #1 (Comments to the Authors (Required)):

The authors have put a lot of effort into revising their manuscript, both in terms of writing and experimentation.

It is now in a much better shape and will be of significant interest in the field.

The authors have responded convincingly to my comments, so I can only recommend publication.

Response: we thank the reviewer for the kind and encouraging words.

Reviewer #2 (Comments to the Authors (Required)):

We thank the authors for their detailed answers and additional work along with the updates made to both the figures and the text.

First, the study documents the existence of two distinct CENP-A nucleosomal fractions in centromere chromatin: CENP-C-bound or not. This work reinforces previous studies (Henikoff et al 2015, Kyriacou & Heun 2018), including earlier work by the authors (Melters et al 2019, 2023). One concern that they addressed relates to possible changes during cell cycle. They have now included a cell cycle analysis of the two CENP-A fractions supporting their existence throughout the cell cycle, while also documenting the structures/complexes associated with each fraction. However, additional controls are missing to confirm cell synchronization in figure 4, such as IF images using EdU to label DNA synthesis and DAPI intensity to evidence enrichment in different cell cycle populations.

Response: We thank the reviewer, as well as reviewer #3, for the suggestion to include evidence for successful cell cycle synchronization. We performed FACS analysis and this data is presented in Supplemental Figure S5.

Second, to investigate the importance of the two distinct CENP-A fractions, the authors overexpressed CENP-C and used CENP-A mutant to "alter the binding landscape of CENP-C" (L318). The authors predict that CENP-A mutants could function as sink for excess CENP-C or increase the kinetochore-free CENP-A population (figure S5). Documenting the functional relevance of these two CENP-A fractions represents a valuable contribution to the field, but additional experiments were missing in order to confirm that CENP-A mutants restore a proper balance between the two CENP-A fractions. The authors now included biochemical and microscopy approaches to examine the CENP-A mutant constructs. However, to make a convincing argument, several issues remain unresolved and further improvement is needed for this critical part of the study as outlined below:

Response: We thank the reviewer for these comments as well as the suggestions made below. For detailed responses, we kindly refer to each individual point.

(1) Biochemical analyses in figure 6 indicate that the HA-H3CENP-A-CTD does not pull-down CENP-C. In addition, incorporation into chromatin of CENP-A mutants remains questionable since neither H2A nor endogenous CENP-A co-IP with them. Therefore, there is doubt concerning

functionality of these mutants and it is difficult to draw a strong conclusion regarding the importance of the balance between the two CENP-A fractions in chromatin.

Response: We thank the reviewer for this comment. In our hands, different CENP-A mutants have very different phenotypes, especially when bound to a tag. This led us last year to publish a paper to address the impact of tags on CENP-A localization (see Bui *et al* 2024 Epigenetics & Chromatin). In this paper we didn't see H2A/B by Western blot following nChIP for tagged CENP-A (Figures 7B and Suppl Figure S3). Yet, by AFM, ChIP-seq, electrophoresis following MNase, and IF, we observed nucleosome sized particles, mononucleosomes sized DNA fragments, and colocalization with the centromere marker CENP-C. In this manuscript, we performed AFM and IF analyses and found nucleosome-sized particles (Figure 6B) and colocalization with centromere markers (Figure 7D, Supplemental Figure S6). We are therefore confident that both CENP-A mutants form nucleosomes. One possibility is that tagged CENP-A mutant nucleosomes obtain different PTMs compared to native untagged CENP-A nucleosomes. Testing this hypothesis goes beyond the scope of this manuscript.

(2) Although the analysis suggests a trend, the CENP-A mutants do not restore significant centromere transcription, as shown in figure 7B, which contradicts the conclusion "CENP-A mutants restore centromeric transcription" (L356, L392...)

Response: We thank the reviewer for this comment. As can be seen in Figure 7B, in the CENP-C overexpression background, both CENP-A mutants do restore centromeric a-satellite transcription to levels indistinguishable from WT as indicated by the *p*-value of 0.938 and 0.758, respectively.

(3) Figure 7C-D requires further attention. Percentages should be indicated in the bar plot, and a statistical analysis is missing in the analysis.

Response: We thank the reviewer for this comment. We have added percentages as requested.

(4) Finally, the authors adapted their model, now proposing that "CENP-C is critical in regulating centromere homeostasis by supporting a unique higher-order structure of centromeric chromatin and altering the accessibility of the centromeric chromatin fiber for transcriptional machinery" (L26...). This conclusion has already been proposed in earlier work by the authors (Melters *et al* 2023), thus the novelty of the contribution is now lost.

Response: We thank the reviewer for this comment. The data presented in this manuscript provide the first fully characterized presence of two CENP-A populations, whereas previous work, including our own, has hinted at the possibility. The full characterization includes previously unrecognized physical nucleosomal difference between the two CENP-A population and the dimension of the CENP-C complex across the cell cycle. In addition, the CENP-A mutant experiments provide novel insights in how centromeric transcription and multipolar spindle errors are regulated.

Other concerns:

- Fig. 3: Immuno-AFM targeting CENP-A indicates the presence of CENP-A in CENP-C complexes. In the text, the authors conclude about the "identity of the nucleosomes associated with CENP-C" (L179, L181, L1054...). However, the results do not support this conclusion and rather indicates the presence of CENP-A nucleosomes.

Response: We thank the reviewer for this comment. As the reviewer can appreciate, we state in lines 204-6 "These results support the notion that CENP-A nucleosomes are indeed associated with the purified CENP-C complexes.", acknowledging the presence of CENP-A nucleosomes associated with CENP-C complexes.

- For similar reasons, we recommend avoiding the use of the term "CENP-A ChIP" to describe the ACA nChIP which targets CENP-A, CENP-B and CENP-C (both in the text and figure legends).

Response: We thank the reviewer for this comment. The text correctly differentiates whether a 2nd ACA nChIP or the 2nd CENP-A nChIP was performed. In each figure, the precise antibody/serum

used for any nChIP is shown per experiment. For clarity to the reader and the figure legends reflect this.

- Fig. 6A: the mCherry tag should be displayed on the scheme.

Response: we have added the mCherry tagged CENP-A to Figure 6A as suggested.

- Fig. 6D: A CSK extraction prior to imaging could provide a good indication to determine if CENP-A mutants can get incorporated into chromatin (with respect to point 1).

Response: We thank the reviewer for pointing out this technical detail. All our IF imaging were performed using CSK extraction. We have updated the methods section to correctly reflect this.

- Fig. 6D: Centromere localization should be controlled through CENP-B or CREST staining.

Response: We followed up on this suggestion and transfected HeLa cells with GFP-tagged CENP-A mutants and co-stained for GFP and endogenous CENP-C (centromere marker). Using the ImageJ colocalization feature, we show that both CENP-A mutants colocalize with CENP-C. This data is presented in Figure 6D.

- Fig. 6D: The DAPI is in blue and the GFP construct in green. Thus, in the top panel, the merged image shows a light, green-colored nucleus (mix of DAPI + mislocalized HA-H3CENP-A-CTD everywhere in the nucleus). However, in the bottom panel, the nucleus is also green (although GFP-CENP-AdCTD appears as green dots on the left image). Are the parameters and range of displayed pixel values the same between conditions?

Response: We thank the reviewer for this observation. The parameters and range used to display the IF images in Figure 6D (now Supplemental Figure S6) are all the same. The same parameters were also used for new Figures 6D and Figure 7C.

- The title of figure 6 (L1066) is incorrect since the HA-H3CENP-A-CTD mutant does not localize to the centromere (as expected).

Response: We thank the reviewer for this comment. We have updated Figure 6D (and the old Figure 6D is now Supplemental Figure S6) to include the use of an endogenous centromere marker CENP-C. We do observe that H3[^]CENP-A CTD goes to both the centromere and to non-centromeric regions, based on IF data (Figure 6D, Supplemental Figure S6). We agree with the reviewer that the global incorporation of H3[^]CENP-A CTD observation is expected as we predicted in Supplemental Figure S7. For this manuscript the important observation is that H3[^]CENP-A CTD also colocalizes with mCherry-CENP-A and endogenous CENP-C.

- L36-37: The two-layered structure of the kinetochore observed by electron microscopy was first showed in Robbins & Gonatas 1964 and described in Brinkley & Stubblefield 1966.

Response: We thank the reviewer for making us aware of these two papers. We have included them in the manuscript.

- L42: CENP-A is not the "foundation" for CENP-B binding, whose targeting is driven by a specific DNA sequence (Masumoto et al 1989) independently from the nucleosomes present (Carroll et al 2009).

Response: We thank the reviewer for this detail. We have updated the text to more accurately reflect the link between CENP-A, CENP-C, and CENP-N.

- In L352, although CENP-A mutants form nucleosome sized particles and colocalize with mCherry-CENP-A, the conclusion "CENP-A mutants appear to form nucleosomes" is not sufficiently substantiated (with respect to point 1).

Response: To further support our conclusion that CENP-A mutants appear to form nucleosomes, we now include IF experiments following CSK extraction where we used the centromere marker CENP-

C. This data is now presented in Figure 6D. We also like to refer our response to major point 1 for further clarification why we are confident that both CENP-A mutants appear to form nucleosomes.

- Fig. 3 legend: how many assays and samples?

Response: the number of particles measured is reported in the figure per condition, similar to all other AFM data presented in Figures 2, 4, 5, and 6. In each legend, we have clarified how many independent experiments were performed.

Minor formatting issues:

- L71: "a CENP-C complex" or "CENP-A complexes"

Response: updated

- L80: "from one another"

Response: updated

- L80: "the potential role of these two"

Response: updated

- L91: "In previous work (...), we noticed"

Response: updated

- L209: Regarding point 2, we would suggest changing the title (for example for "CENP-A nucleosomes associated with CENP-C are physically distinct")

Response: updated as suggested

- L221 ("we examined nucleosomes that are physically associated with the CENP-C complexes") is redundant with L210 ("we next turned our attention to individual nucleosomes associated with the CENP-C complex"), making it hard to follow/understand the objectives.

Response: we updated the text to resolve this redundancy.

- L318: "we reasoned that introducing CENP-A mutants would alter"

Response: updated

- L368: "First, we performed"

Response: updated

- L402: "the most noticeable"

Response: updated

- L430: "Only one"

Response: updated

- L520: "were driven ... and were made"

Response: updated

- L612: "after treating the samples with proteinase K"

Response: updated

- L1024: "to the CENP-C complex"

Response: updated

- L1042: "done on the unbound fraction"

Response: updated

- L1072: "and the diameter of the mutants was determined using"

Response: updated

- Fig. 2B: The p-value is hidden behind a plot

Response: this was for Figure 3B. The error has been corrected.

- Fig. 3 legend: (A), (B) and (C) are misplaced, and (D) is missing

Response: We have updated the locations of Figure 3 legends for A, B, and C. There is no Figure 3D.

- Fig. 6 legend: the first sentence is unclear

Response: We have updated Figure 6 legend for clarity.

Reviewer #3 (Comments to the Authors (Required)):

This manuscript has been substantially revised and focusses on the key technical advances. There is a significant improvement in how the manuscript flows and the layout of the figures. The authors took our advice to look at cell cycle stage and I believe this paid off. Whilst I am generally satisfied and I appreciate the effort the authors have put in to improve this manuscript since first submission, there lacks a key control for cell cycle stage. I cannot accept this manuscript without properly controlling for this. The timings suggested in Fig 4A look correct, but cell cycle stage shouldn't be assumed. The timings need to be shown to enrich these cell cycle stages. This can be shown by PI staining (DNA content) and flow cytometry, for example. i.e. what fraction of cells are enriched for each cell cycle stage at each time point? I am happy to accept pending this important addition.

Response: we thank the reviewer for their kind and encouraging words. We have added FACS analysis as a control for the cell cycle stage selection following the double thymidine block. This data can be found in Supplemental Figure S5, and the manuscript has been updated accordingly.

Minor points:

1) Fig 1B: I recommend updating the blot for H2B here. Whilst it probably doesn't change the conclusion of the figure, it is worth getting a cleaner blot for H2B here given it is a main figure. It looks overexposed and the 2nd NChIP lane has a poor H2B band.

Response: We have updated the H2B lane as requested by the reviewer.

2) I would possibly suggest revising the CpC and CpA to simply CENP-A and CENP-A.

Response: we tried to convert CpA/C to CENP-A/C where possible while maintaining a uniform and consistent format and font size across all figures. We have updated the manuscript and figures to only use CENP-A and CENP-C instead of CpC and CpA.

January 13, 2025

RE: Life Science Alliance Manuscript #LSA-2024-02819RR

Dr. Yamini P Dalal
National Institutes of Health
National Cancer Inst. (NCI)
41 Library Drive MSC 5055
Bethesda, MD 20892-5055

Dear Dr. Dalal,

Thank you for submitting your revised manuscript entitled "High-Resolution Analysis of Human Centromeric Chromatin". We would be happy to publish your paper in Life Science Alliance pending final revisions necessary to meet our formatting guidelines.

- please be sure that the authorship listing and order is correct
- please add a Data Availability statement at the end of the Materials and Methods section to highlight the accession information for the ChIP-seq dataset
- please add sizes next to all blots

A. FINAL FILES:

B. MANUSCRIPT ORGANIZATION AND FORMATTING:

Sincerely,

January 16, 2025

RE: Life Science Alliance Manuscript #LSA-2024-02819RRR

Dr. Yamini P Dalal
National Institutes of Health
National Cancer Inst. (NCI)
41 Library Drive MSC 5055
Bethesda, MD 20892-5055

Dear Dr. Dalal,

Thank you for submitting your Research Article entitled "High-Resolution Analysis of Human Centromeric Chromatin". It is a pleasure to let you know that your manuscript is now accepted for publication in Life Science Alliance. Congratulations on this interesting work.

DISTRIBUTION OF MATERIALS:

Again, congratulations on a very nice paper. I hope you found the review process to be constructive and are pleased with how the manuscript was handled editorially. We look forward to future exciting submissions from your lab.

Sincerely,
